# How Text Quality Interventions Reshape Neural Scaling Laws for LLMs: Empirical Study

**Newsha Ardalani[1][†], Feiyang Kang[1,2,\*], Michael Kuchnik[1], Mostafa Elhoushi[3,\*],**
**Shubho Sengupta[4,\*], Shang-Wen Li[1], Carole-Jean Wu[1]**

[1]FAIR at Meta, [2]Virginia Tech, [3]Cerebras Systems, [4]Axiom Math

## Abstract

Neural scaling laws are widely used for performance projection and resource planning, yet their sensitivity to data quality interventions remains poorly understood. We present the first large-scale empirical study of how interventions—deduplication, heuristic filtering, and LLM-guided rewriting—reshape scaling behavior in large language model training. Using QualityPajama, a suite of 23 systematically curated datasets, we train over 2,000 models (100M–8B parameters, 100M–200B tokens) to measure how text quality interventions affects scaling-law parameters and compute-optimal design decisions. While prior studies have shown that model architecture primarily shifts coefficients, we demonstrate that data interventions shift both coefficients and exponents, fundamentally changing the fitted scaling laws in ways not anticipated by existing theory. We show that data quality ranking is scale and resource-dependent. Compute-optimal token–to-parameter ratios vary by orders of magnitude across interventions, revealing a fundamental data quality–quantity trade-off in scaling. These findings pave the way for deeper theoretical understanding of scaling laws, establish scaling-law analysis as a principled framework for data strategy evaluation and ranking, and motivate a data-quality–aware approach to scaling LLMs.

## 1 Introduction

While nearly all large language models are trained on similar sources of text—web data— the key differentiating factor among state-of-the-art models lies in the quality of their pre-training and post-training data. However, data quality itself remains an elusive and context-dependent concept—what constitutes "high quality" can vary with downstream use case, compute scale, and resource constraints. This raises the question: *can neural scaling laws offer a principled framework for ranking data quality across scales?*

Neural scaling laws are empirical relationships that describe how model performance improves as a function of resource investment - typically the number of parameters and training tokens. A growing body of empirical (Hestness et al., 2017; Johnson & Nguyen, 2017; Rosenfeld et al., 2019; Kaplan et al., 2020; Hernandez et al., 2021; Ghorbani et al., 2021; Ardalani et al., 2022; Hoffmann et al., 2022; Alabdulmohsin et al., 2022; Aghajanyan et al., 2023; Isik et al., 2024; Zhang et al., 2024) and theoretical (Sharma & Kaplan, 2022; Bahri et al., 2024; Brill, 2024; Hutter, 2021; Michaud et al., 2023; Dohmatob et al., 2024b; Dębowski, 2023; Dohmatob et al., 2024a) work has shown that pre-training loss follows a power-law trend with respect to these axes. Neural scaling laws have been central to the development of large language models (LLMs), informing decisions about model scaling, data scaling, and compute allocation, while also serving as a key tool for return-on-investment (ROI) analysis and capability forecasting (Hestness et al., 2019; Hoiem et al., 2021; Mahmood et al., 2022; Alabdulmohsin et al., 2022). However, despite their widespread adoption, the impact of data quality on scaling laws remains poorly understood.

A prominent example of this uncertainty is the ongoing debate over the discrepancy between Kaplan et al. (2020)'s and Hoffmann et al. (2022)'s prediction of the compute-optimal token-to-parameter

---

\*Work done at Meta. [†] Correspondence to: Newsha Ardalani <new@meta.com>.

ratio (21 vs. 1) (Porian et al., 2024; Pearce & Song, 2024; Bi et al., 2024). Recent work speculates that differences in training data may have played a role in this divergence (Bi et al., 2024). While prior theoretical works (Sharma & Kaplan, 2022; Bahri et al., 2024; Brill, 2024; Hutter, 2021; Michaud et al., 2023; Dohmatob et al., 2024b; Dębowski, 2023; Dohmatob et al., 2024a) have linked the power-law *exponents* to properties of the data manifold and the Zipfian distribution of input tokens, the impact of *text quality* interventions on these underlying structures remains poorly understood. Furthermore, prior work overlooks how data quality influences **other components** of the scaling law—namely, the coefficients and asymptotic loss terms—which, as we will show, play a critical role in shaping loss behavior at today's compute scales. Moreover, most theoretical predictions isolate a single exponent (either model or data) while holding the other in the infinite limit. As we demonstrate, understanding the **joint fit** is essential, as the components often move in opposing directions to control loss trajectory, revealing important trade-offs induced by data quality shifts. Although prior empirical work has explored the effects of synthetic noise, data source composition, and filtering algorithms in domains such as machine translation (Bansal et al., 2022) and image classification (Bahri et al., 2024), to the best of our knowledge, there is no systematic study examining how *text-specific interventions*—such as filtering, deduplication, rephrasing and mixing synthetic and natural data—impact the components of neural scaling laws in LLM pretraining.

Our work bridges this gap by conducting a large-scale empirical analysis of diverse data quality interventions for pretraining large scale language models and study how they influence **all** components of the scaling law. We introduce a benchmark of 23 curated datasets, each representing a different quality intervention, and train over 100 language models per dataset, totaling more than 2000 model training runs. This extensive experimental design enables us to disentangle the effects of data quality on scaling law components and loss behavior, and propose how to design an effective data quality strategy as we scale.

## 1.1 OUR CONTRIBUTIONS

- **QualityPajama Benchmark:** We introduce *QualityPajama*, a benchmark suite of 23 datasets designed to systematically evaluate the impact of diverse text quality interventions on neural scaling behavior in LLMs. (Section 3)

- **Full Scaling Law Decomposition:** We provide the first systematic analysis of how text-quality interventions affect all components of the joint scaling law—not only the exponents. Our results show that stronger filtering does not consistently push components toward more favorable regimes, but instead produces conflicting shifts across parameters. (Section 4)

- **Data-Aware Scaling Strategies:** We show that designing compute-optimal scaling strategies requires careful accounting for data quality, as variation in quality could shift the optimal number of parameters, tokens, and their ratio by couple orders of magnitude.(Section 4.1)

- **Scale- and Resource-Dependent Rankings:** Data quality rankings are not uniform across scales or resource regimes. Strategies that excel at small scales may underperform at larger ones, and the optimal choice depends critically on the constraint (e.g., fixed compute vs. fixed data). Moreover, "scale" can refer to model size, dataset size, or compute budget, and the best intervention differs across these regimes. We recommend using scaling-law curves to rank data quality strategies across different scales and resource constraints, rather than relying on small-scale experiments, which often lead to misleading conclusions. (Section 4.1)

- **Deduplication Efficiency:** We demonstrate that deduplication yields large compute savings that far exceed reductions in data volume (Section 5)

- **PageRank Signals:** While PageRank scores correlate with improved quality, filtering based solely on PageRank does not outperform the unfiltered baseline. (Section 5)

- **Synthetic–Natural Data Mixing:** We show that mixing synthetic and natural data consistently outperforms using either alone, but the optimal mixing ratio evolves as the model and compute scale. (Section 5)

## 2 BACKGROUND AND RELATED WORK

The study of scaling laws in deep learning has a rich history, with numerous empirical (Hestness et al., 2017; Johnson & Nguyen, 2017; Rosenfeld et al., 2019; Kaplan et al., 2020; Hernandez et al., 2021; Ghorbani et al., 2021; Ardalani et al., 2022; Hoffmann et al., 2022; Alabdulmohsin et al., 2022; Aghajanyan et al., 2023; Isik et al., 2024; Zhang et al., 2024) and theoretical Sharma & Kaplan (2022); Bahri et al. (2024); Brill (2024); Hutter (2021); Michaud et al. (2023); Dohmatob et al. (2024b); Dębowski (2023); Dohmatob et al. (2024a) investigations into their components. A commonly used form of the scaling law is given by:

$$\text{Loss}(N, D) \sim AD^{-\alpha} + BN^{-\beta} + E$$

where Loss typically represents cross-entropy loss, $D$ denotes data size in tokens, $N$ represents model size in parameters, and $\alpha$, $\beta$, $A$, $B$, and $E$ are constants. The terms in this equation capture the effects of finite data, limited model capacity, and the inherent entropy of the underlying phenomenon, respectively.

Although prior work has empirically explored the impact of model architecture (Tay et al., 2022), vocabulary size and tokenizer on the components of scaling law (Hestness et al., 2017; Kaplan et al., 2020), the impact of data quality on all components of scaling law remains poorly understood. Prior theoretical works on the origin of the power law and its relation to the dimensionality of data manifold (Sharma & Kaplan, 2022; Bahri et al., 2024) and Zipfian distribution of input data (Hutter, 2021; Michaud et al., 2023) are perhaps closest to our own. usually under some simplifying assumptions like infinite data size or model size. They particularly make predictions about the exponents of power law but remain silent about other components.

**Data Manifold Theory:** Data manifold refers to the low-dimensional structure that higher dimensional data lies on. Data manifold theory predicts that exponents of power law are inversely proportional to the data manifold dimension (Sharma & Kaplan, 2022; Bahri et al., 2024). However, the impact of data quality on data manifold itself is poorly understood. Data quality, particularly text quality, can be characterized across various axes: diversity of topics, grammar complexity, formatting artifacts, information density, factuality, fairness, safety, etc. While prior theoretical work do not discuss the impact of data quality explicitly, their machinery is powerful enough to make predictions. Take removing unstructured noise, like garbled text, it could ostensibly decrease the apparent dimensionality. On the other hand, deduplication could expand the data manifold. While both are different text interventions towards improving quality, one seems to improve the exponent, while the other decreases.

**Zipfian Distribution Theory:** Zipf's law is another empirical observation that explains word frequencies follow a power-law in their rank. It shows up not only in word frequencies, but also in n-gram distributions (Ha et al., 2009), sentence structures, and higher-level concepts (Michaud et al., 2023). Prior work conjectures that if input data follows a Zipfian distribution, the Zipf's exponent correlates with the power law exponent (Hutter, 2021; Michaud et al., 2023). However, much like data manifold theory, the impact of data quality interventions on Zipfian distribution are not quite predictable. While some data intervention techniques, like synthetic data generation cuts off the heavy tail of the input distribution, other intervention techniques like deduplication flattens the head of the curve. This implies that Zipfian slope gets steeper for synthetic data but flatter for deduplicated data. We will see later in Section 4, these predictions are not always consistent with empirical observations as it is not easy to predict how data quality interventions influence distribution.

**Effective Tokens and Utility-Based Scaling Laws:** Prior work has examined how to incorporate data quality into scaling law formulations. Chang et al. (2024) focus only on the data axis, proposing to replace dataset size $D$ with an *effective* variant, but leaving other components of the law unchanged. Muennighoff et al. (2023) extend this idea to both model size and dataset size, introducing effective formulations $N'$ and $D'$, though their analysis is tailored to the setting of repeated epoching rather than data interventions. Goyal et al. (2024) similarly reinterpret the data exponent $\beta$ in terms of *effective utility*. These approaches capture aspects of data efficiency but treat quality as primarily modifying $D$ or $\beta$, overlooking its broader influence on parameter coefficient and exponents, or irreducible loss. By contrast, we show that data interventions perturb *all* components of the joint scaling law fit. Most recently, Shukor et al. (2025) proposed a "full" scaling law for data

mixtures, which is closest in spirit to our work. Their focus is on mixture composition as the intervention, whereas we analyze heuristic filtering and synthetic data rewrites, broadening the range of data-centric interventions studied under scaling laws. Overall, our work is the first to demonstrate that text quality interventions affect *all* components of the scaling law, not just the data dimension, providing a more complete picture of how quality reshapes scaling dynamics and offering practical guidance for data-centric scaling strategies.

**Synthetic Data Scaling Laws**     Fan et al. (2024) studied the impact of synthetic images on scaling laws, particularly on data exponent. Qin et al. (2025) examined how generator model size influences scaling laws on downstream tasks for LLMs. In contrast, we study upstream loss and investigate how mixing synthetic and natural data shapes scaling behavior in LLM.

**Dynamic Data Intervention and Non-Power-Law Scaling**     Sorscher et al. (2022) show that, with adaptive data pruning during training, it is possible to surpass standard power-law scaling and approach exponential improvements. In contrast, our work assumes interventions are applied once prior to pre-training, rather than adaptively adjusting throughout training.

**Post-Training Data Quality**     Recent work has investigated the role of data quality over quantity in post-training alignment, showing that even small high-quality datasets improve performance (Zhou et al., 2023; Xia et al., 2024).

**Text Quality Interventions**     can be characterized across multiple axes, including information entropy, topical diversity, grammar complexity, formatting artifacts, factuality, fairness, and safety. The exact definition of text quality typically varies by downstream usecase. In this work, we focus on the impact of text quality on upstream loss. Broadly, data quality can be manipulated through three strategies: **filtering**, which removes low-quality or undesired content using heuristics or model-based approaches (Raffel et al., 2020; Lee et al., 2021; Sharma et al., 2024); **mixing**, which rebalances data distributions or adds high-quality subsets (Li et al., 2024; Shukor et al., 2025); and **synthetic generation**, which uses LLMs to clean or augment existing content (Kang et al., 2025). These approaches have informed the design of many recent LLM training corpora, including RedPajama (AI, 2023), Dolma (Soldaini et al., 2024), RefinedWeb (Penedo et al., 2023), FineWeb (Penedo et al., 2024), DCLM (Li et al., 2024). Yang et al. (2023) shows how to use a small seed of synthetically corrupted bad data to cluster and filter out real bad data in code.

## 3   QUALITYPAJAMA

We introduce QualityPajama, a benchmark suite of 23 datasets derived from Common Crawl, each reflecting a distinct level of data quality and intervention. The suite spans a broad spectrum of data quality techniques, including 14 filtered datasets and 9 synthetically curated datasets, for training large language models. Table 1 summarizes the interventions used in each category. Additional details regarding dataset construction and design choices can be found in Appendix.

## 4   IMPACT OF DATA QUALITY ON SCALING LAW COMPONENTS

We aim to understand how data quality affects scaling law components, whether predictable patterns emerge under quality interventions, and how these insights can guide effective data curation.

Figure 1 visualize the impact of text quality interventions, particularly heuristic-based filtering and synthetic data generation, on components of neural scaling laws, namely $\alpha$, $\beta$, $A$, $B$ and $E$. Each line in the radial plot represents a different training set, while the radial axis displays various validation sets. It is apparent from these results that all components are sensitive to training set quality as well as validation set quality.

**Sequential Application of Data Filters and Effects on Scaling Components**   We apply a series of data filters sequentially and extract intermediate datasets at each stage to conduct scaling law analysis. The order in which filters are applied is indicated in the legend. Interestingly, the trajectory of changes in scaling law components does not necessarily follow the order of interventions. Take $\alpha$ for example: it increases after removing NSFW content (red to green), but decreases after filtering

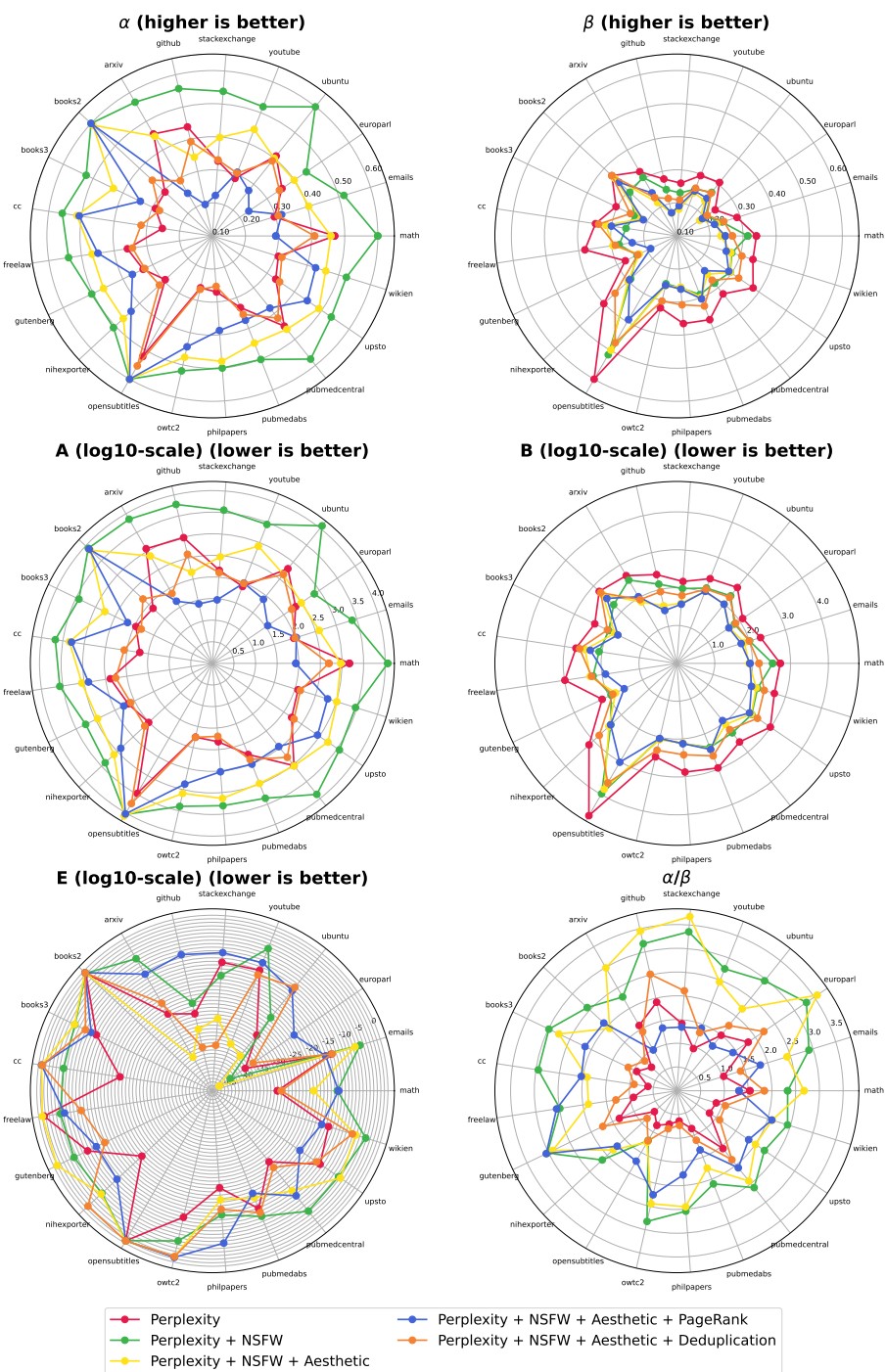

Figure 1: **How Data Filtering Affects Scaling Law Components.** Different colored lines represent different data-quality interventions in training set, while the radial axes show impact across different validation sets. While parameter estimates vary across evaluation domains due to train–test distribution shift, the consistent separation between filter-specific curves across validation sets indicates that data-quality interventions induce systematic shifts in scaling-law parameters.

Table 1: Summary of QualityPajama dataset interventions.

| Category | Description | Abbrev./Variants |
|---|---|---|
| *Heuristic-based Filters (14 variants)* | | |
| NSFW Filtering | Removes documents containing offensive or inappropriate content. | `nsfw` |
| Aesthetic Filters | Filters out text with undesirable patterns (e.g., "lorem ipsum", in-line code, or high alphanumeric ratios > 0.8). | `aesthetic` |
| PageRank Filtering | Partitions pages into low/medium/high/unknown based on PageRank score. Thresholds are set to the 33rd and 67th percentiles of the score distribution of all pages in the PageRank table. (Page et al., 1999). | `high_pr, med_pr, low_pr, no_pr` |
| Deduplication | Fuzzy deduplication using MinHashLSH (Leskovec et al., 2020) with different similarity thresholds; selects lowest perplexity document from near-duplicate clusters. | `deduped_0.7, deduped_0.8, deduped_0.9, deduped_1.0` |
| Grammar Complexity | Filters based on average sentence length as a proxy for syntactic richness, with thresholds at 10 tokens for short text and 25 tokens for medium text | `short_text, medium_text, long_text` |
| *Synthetic Curation (9 variants)* | | |
| High Quality Rephrasing (HQ) | LLM rewrites documents to be clearer and more coherent (Maini et al., 2024). Mixtures denote the percentage of synthetic vs. natural data (CC). | `HQ100, HQ67-CC33, HQ33-CC67.` |
| Question Answering Rephrasing (QA) | LLM converts documents into conversational QA pairs. | `QA100, QA67-CC33, QA33-CC67` |
| Textbook-style Rephrasing (TB) | Converts documents into textbook-style chapters using structured prompting (inspired by Phi models (Li et al., 2023; Javaheripi et al., 2023; Abdin et al., 2024)). | `TB100, TB67-CC33, TB33-CC67` |

Table 2: **Can Scaling Components Reliably Rank Data Interventions?** We report average Spearman correlations across validation sets for each scaling law component. Moderate values (0.3–0.5) suggest that component-based rankings are only partially preserved across validation sets; higher values suggest reliable ordering. Results suggest that such metrics may not reliably rank natural data interventions. In contrast, rankings for synthetically curated datasets show strong consistency, suggesting scaling components are more reliable for evaluating synthetic data strategies.

| Data Interventions | **A** | **B** | $\alpha$ | $\beta$ | **E** |
|---|---|---|---|---|---|
| All heuristic filters | 0.45 | 0.34 | 0.46 | 0.32 | 0.34 |
| All synthetic data | 0.81 | 0.91 | 0.76 | 0.91 | 0.54 |

garbled text (green to yellow). It decreases further after removing pages with low PageRank scores (yellow to blue), but then increases again after deduplication (blue to orange). As shown in Appendix A, these dynamics are not always consistent with predictions from Zipf's law or the data manifold hypothesis, showcasing the limitations of the current theory.

**Component-Wise Correlations** We examine whether scaling law components exhibit consistent patterns under data quality changes—for example, whether improving quality increases the model exponent ($\alpha$) and decreases the data exponent ($\beta$). While Figure 1 suggests such trends qualitatively, we quantify them via Spearman correlations (Figures 2a and 2b). The strongest, most stable correlations across all validation sets are: $A \propto \alpha$ and $B \propto \beta$.

**Sensitivity to Validation Sets** We examine whether data intervention rankings are consistent across validation sets. Table 2 reports average Spearman correlations per component. While filters in Figure 1 show high consistency, rankings across all 14 heuristic filters exhibit only moderate correlation (0.3–0.5), suggesting that filter rankings are only partially preserved across validation sets. **This indicates that scaling law behavior is not independent of the validation set for *naturally* curated datasets.** On the contrary, for synthetically curated datasets, we see a strong correlation across validation sets. **This indicates that scaling law behavior is less sensitive to validation set for *synthetically* curated datasets.**

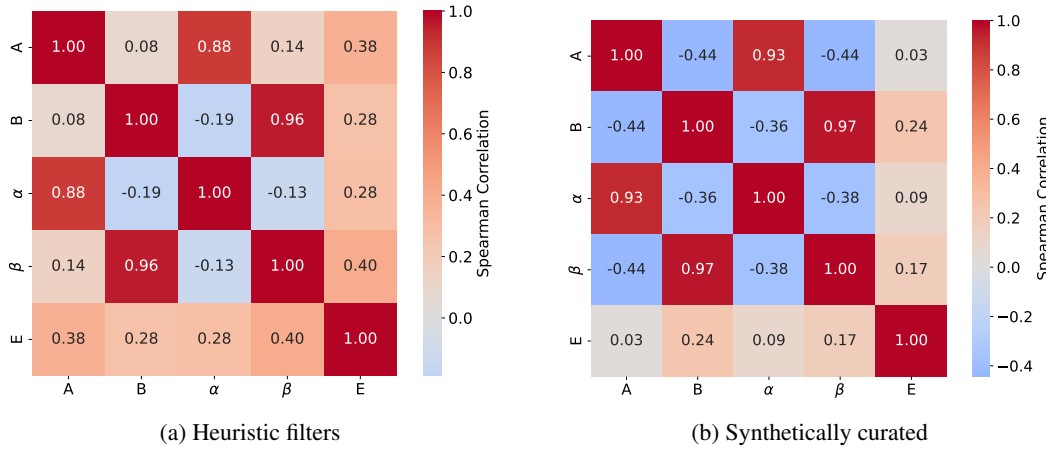

Figure 2: **How Scaling Law Components Co-Vary with Data Quality Intervention?**. We observe strong monotonic correlations between $A$ and $\alpha$, and between $B$ and $\beta$. For synthetic data, there are also notable negative correlations between $\alpha$ and $\beta$, and between $A$ and $B$.

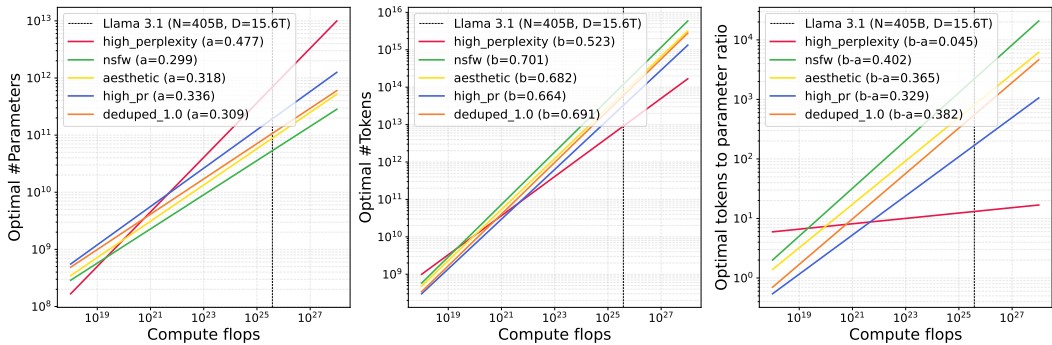

Figure 3: **How Data Quality Influences Scaling Strategy?** Given $N_{\text{opt}} \propto C^a$, $D_{\text{opt}} \propto C^b$, and $D_{\text{opt}}/N_{\text{opt}} \propto C^{b-a}$, where $a = \beta/(\alpha + \beta)$ and $b = \alpha/(\alpha + \beta)$. **(Left)** shows how optimal model size scales with compute. At today's compute budget (dashed line), the best and worst data interventions differ by over an order of magnitude in optimal model size. **(Right)** shows the variation in token-to-parameter ratio, where interventions differ by up to two orders of magnitude at the same compute scale.

## 4.1 INTERPRETATIONS

**Designing Compute-Optimal Scaling Strategy Requires Accounting for Data Quality:** Prior work has shown that compute-optimal design decisions depend on the scaling law components: $\alpha$, $\beta$, $A$, and $B$. Since data quality influences these parameters, it directly affects compute-optimal choices. Figure 3 illustrates how the compute-optimal number of tokens, number of parameters, and their respective ratio (a proxy for sample efficiency) scale with available compute and vary with data quality intervention. Notably, at today's compute scale (indicated by the dashed line), the optimal design point can differ significantly across—by up to $14\times$ for the number of parameters, $13\times$ for the number of tokens, and an astonishing $182\times$ for the token-to-parameter ratio. These results highlight the critical role of data quality in determining efficient scaling strategies, underscoring the need to account for quality variations when designing large-scale training runs.

**Tension Among Scaling Law Components:** Data interventions do not uniformly shift all components of the scaling law in a direction that reduces loss. We observe that the coefficients $A$ and $B$ are positively correlated with their corresponding exponents $\alpha$ and $\beta$, respectively. This coupling creates a tension in how different components influence performance. While increasing the expo-

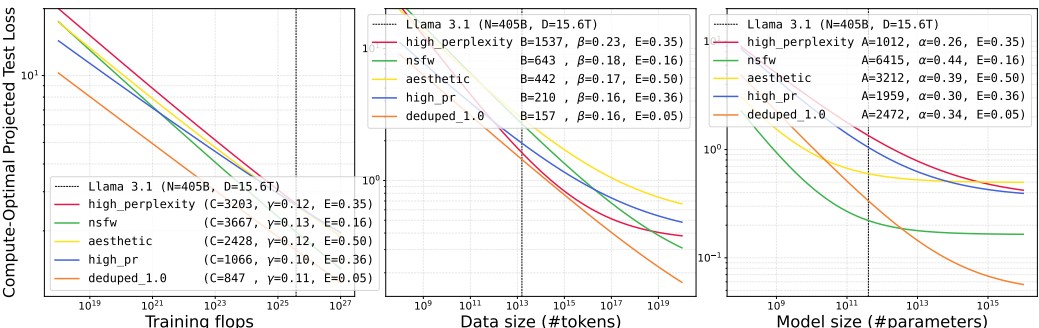

Figure 4: **How Does the Optimal Data Quality Strategy Change with Scale and Resource Constraints?** Compute-scaling law (left), data-scaling law (middle), and model-scaling law (right) curves show that no single data strategy remains optimal across all scales. The optimal choice shifts as the resource scale changes and also depends on which resource is constrained: model size, data size, or compute budget.

nents $\alpha$ and $\beta$ typically leads to improved scaling and lower loss, increases in the coefficients $A$ and $B$ have the opposite effect, raising the loss. As a result, interventions that improve one component may simultaneously degrade another.

One may argue that in the trade-off between exponent and coefficient, the exponent should dominate, since its effect is exponential while the coefficient scales only linearly. While this may hold asymptotically at extremely large scale, Figure 4 shows that the tension persists even at today's compute scale (e.g., $10^{24}$ FLOPs). This persistence may be due to the fact that the coefficients $A$ and $B$ vary across several orders of magnitude, while the exponents $\alpha$ and $\beta$ remain relatively small, limiting their ability to compensate.

Notably, in synthetically curated datasets, we observe a negative correlation between $\alpha$ and $\beta$, suggesting that improvements in model scaling efficiency may come at the expense of data scaling efficiency. Such opposing forces highlight the complex and sometimes counteractive nature of data quality interventions on loss behavior. This underscores the need to analyze all components of the scaling law jointly, rather than relying on any single metric to assess data quality improvements.

**Data Quality Rankings Vary with Scale** We observe frequent crossovers between scaling curves for different data interventions (Figure 4), indicating that a dataset which minimizes loss at small scale may be outperformed by another at larger scale. This shift in relative performance highlights the risk of extrapolating small-scale experimental results to large-scale settings. Consequently, conclusions drawn from limited-scale experiments may not generalize to high-compute regimes, and data quality strategies should be validated at or near the intended scale of deployment to ensure their effectiveness holds under real-world training budgets.

**The Best Data Quality Strategy Depends on Your Resource Constraint** In addition to being scale-dependent, the "best" data quality strategy depends on the specific resource constraint, as shown in Figure 4. For instance, if the goal is to identify the most efficient dataset under a fixed compute budget, compute scaling provides the most relevant lens. However, if the constraint lies in model size or available training tokens, the conclusions may differ. Therefore, practitioners should be mindful of their primary resource constraint when evaluating or selecting data quality strategies, as the optimal choice is inherently constraint-dependent.

## 5 Data Quality Intervention Comparisons Through Compute-Efficiency Lens

How aggressive should deduplication be? Is there a diminishing return in compute efficiency as we dudupe more aggressively? Is PageRank a useful signal for filtering? Is it more or less compute effi-

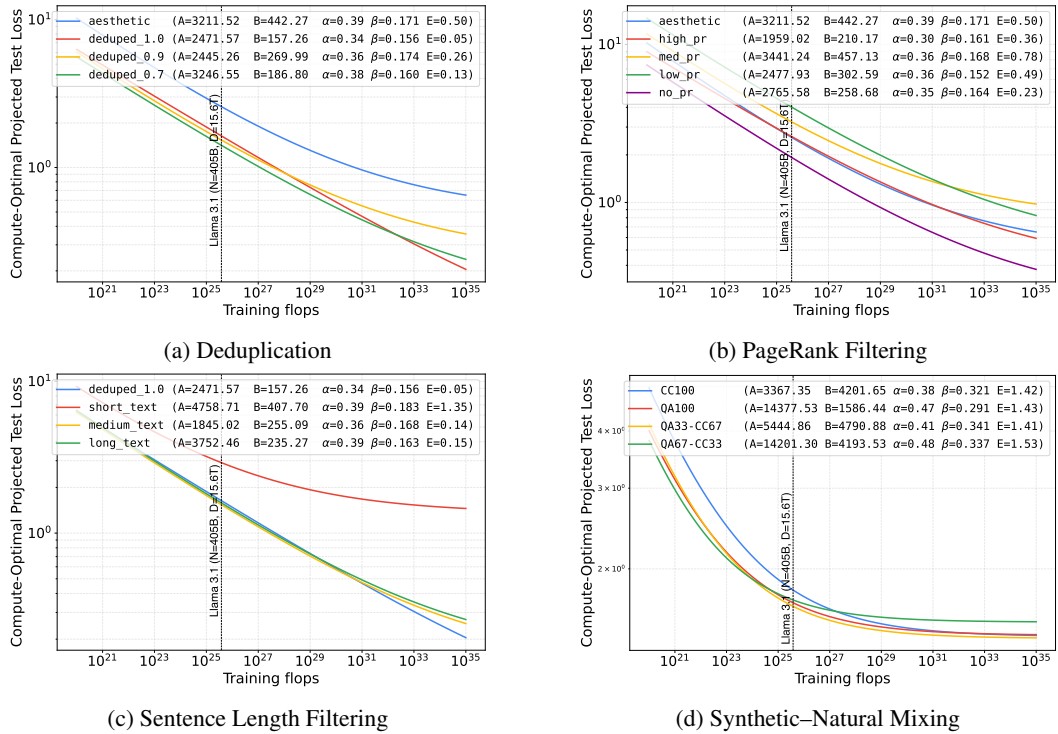

Figure 5: Compute scaling law results for various data quality interventions.

cient to train with synthetic data? Do improvements in compute-efficiency merely reflect reductions in data volume, or can they go further? To address these questions, we analyze compute scaling laws under various data quality interventions in Figure 5.

- **Deduplication:** Fuzzy deduplication offers substantial compute savings that far exceed reductions in dataset size. For example, exact deduplication reduces data volume to 83% of its original size yet yields a 100× gain in compute efficiency. Fuzzier approaches perform even better: `dedupe_0.7` requires approximately 3× less compute than `dedupe_0.9`, 10× less than exact deduplication, and 300× less than no deduplication (Figure 5a).

- **PageRank Filtering:** While a higher PageRank correlates with improved quality (`high_pr` > `med_pr` > `low_pr`), filtering strictly by high PageRank does not outperform the baseline. In contrast, including pages not found in the ranking table (`no_pr`) results in significantly greater compute efficiency—likely due to recency effects (Figure 5b).

- **Synthetic–Natural Mixing:** Mixing synthetic and natural data consistently outperforms using either alone, but the optimal mixing ratio evolves with compute scale (Figure 5d).

## 6 DISCUSSION

**Summary:** We set out to analyze the impact of text quality interventions, particularly heuristic-based filtering and LLM-guided data rewrite, on the components of neural scaling laws in training large language models. To enable this study, we developed QualityPajama, a benchmark suite of 23 systematically constructed text datasets spanning a range of quality levels and interventions, from filtering to deduplication to paraphrasing and synthetic curation built on top of Common Crawl dataset. We found that: (1) all components of the scaling law are sensitive to data quality (2) data intervention rankings are not preserved across scales; (3) the decision on how to scale model size and data size with increased compute budget is heavily influenced by data quality; (4) data intervention impact on compute saving goes far beyond the reduction in data volume; and (5) mixing synthetic and natural data outperforms using either alone, though the optimal ratio is scale dependent.

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

# Appendices

# A TRAINING DATASET

## A.1 BASELINE DATASET CHOICE

We build QualityPajama on top of CommonCrawl dataset assembled by **RedPajama-v2** (AI, 2023), which includes 84 Common Crawl snapshots from 2014 to 2023. RedPajama shares raw Common-Crawl dataset along with quality signals for each document but does not filter out any data from the mix. There is roughly 0.5 TB or 100 B tokens per snapshot per partition. We focus on the English subset from 34 snapshots and head partition, totaling approximately 15 TB of data (or 3 T tokens). This choice is motivated by three key considerations:

- **Minimal Pre-processing:** To be able to evaluate the impact of data quality interventions, we require a dataset that is minimally processed. RedPajama's CommonCrawl is preserving much of its original form while offering a clean interface.

- **Scale:** A dataset of substantial size is necessary to support scaling law analyses across multiple orders of magnitude—even after aggressive filtering. RedPajama-v2, is well suited in terms of both volume and temporal coverage. Given that our final dataset is $\approx 1\%$ of the original dataset, to enable an equal scaling range for all datasets, say upto 30B tokens, the original dataset should be in 3T tokens/15TB range.

- **URL Availability:** The presence of a URL for each document allows us to explore PageRank-based filtering techniques. This is particularly useful given that crawling algorithms like Hyper Centrality (used by Common Crawl) already introduce implicit biases that we can now systematically study.

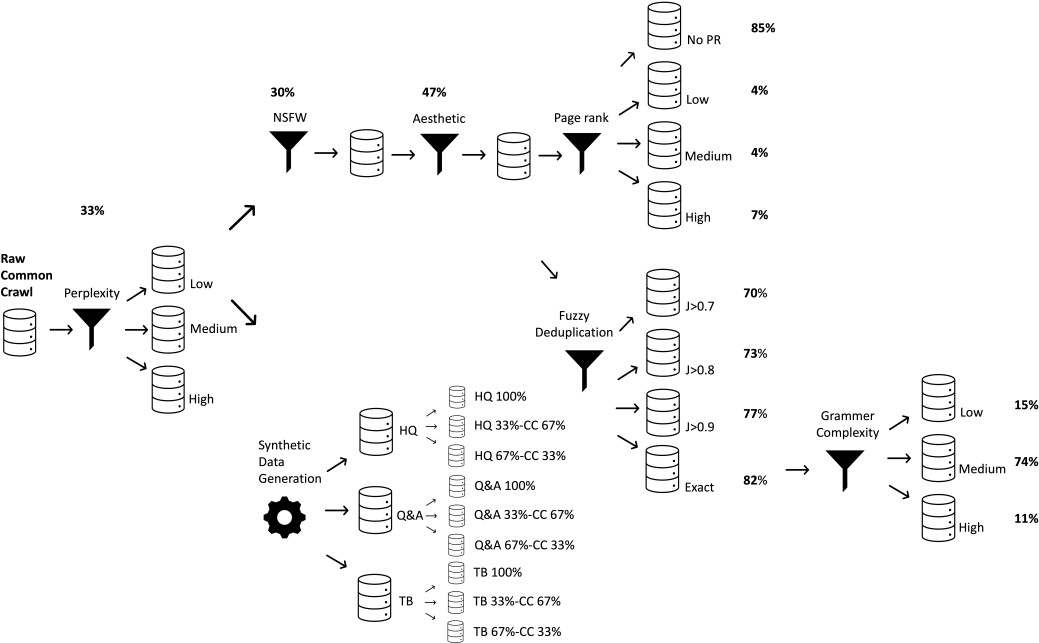

Figure 6: QualityPajama data pipeline. We show the filtering rate next to each filter. Given that our final dataset is $\approx 1\%$ of the original dataset in volume, to enable an equal scaling range for all datasets, say upto 30B tokens, the original dataset should be in 3T tokens/15TB range.

## A.2 DERIVATIVE DATASETS

Figure 6 illustrates the pipeline used to construct the QualityPajama benchmark suite. To support this, we developed a scalable Spark-based (Zaharia et al., 2010) data processing framework—*PajamaKit*—that enables rapid experimentation with filtering, deduplication, and other data curation strategies.

### A.2.1 Heuristic-based Data Quality Filters

We carefully hand-pick a set of filters that are deemed to improve quality to the extent that they are included in many data recipes used for curating well-known datasets such as C4 (Raffel et al., 2020), Dolma (Soldaini et al., 2024), RedPajama (Weber et al., 2024), RefinedWeb (Penedo et al., 2023) and FineWeb (Penedo et al., 2024). These include NSFW filtering, format-based filtering, grammar-based filtering, deduplication, etc. We also include some less explored filters, like PageRanking score to study their effectiveness. We apply these filters sequentially and extract intermediate datasets after each stage. Heuristic filters usually are accompanied with some knobs to control their filtering degree. For instance, deduplication has a similarity threshold for deeming two samples duplicate and we are curious to understand: how does this knob controls quality? Where applicable, we experiment with multiple thresholds and retain only the "best" filtered dataset for downstream filtering. The filtering pipeline includes:

- **NSFW Filtering:** We remove all pages containing inappropriate or offensive language.

- **Aesthetic Filters:** We exclude documents containing undesirable patterns such as "lorem ipsum," inline code (e.g., "{", "javascript"), and those with a high alphanumeric character ratio (above 0.8).

- **PageRank Filtering:** We partition documents into four groups—low, medium, high, and not-found—based on their PageRank scores (Page et al., 1999). Since Common Crawl sampling is biased towards high Hyper Centrality (correlated with PageRank), our analysis exposes implicit biases in many web-derived corpora. The thresholds are chosen to split the PageRank score distribution in our reference table into three equal parts.

- **Deduplication:** We apply deduplication at page granularity within each snapshot. For *fuzzy deduplication*, we use MinHashLSH (Leskovec et al., 2020) at different Jaccard similarity thresholds (0.7, 0.8, 0.9, 1.0). We build MinHash signatures on top of pre-processed lower-cased bi-grams with 256 permutations. We use signature to build a similarity graph, from which connected components (clusters of near-duplicates) are identified. Within each cluster, the document with lowest perplexity score is retained.

- **Grammar Complexity:** We use *average sentence length* as a simple first-order proxy for syntactic complexity. Using NLTK for sentence and token segmentation, we bin documents into categories of short, medium, long, and very long sentences.

### A.2.2 Synthetic Curation Techniques

While the literature on synthetic data generation is very rich, only a few have been proposed and deployed for pretraining large language models (Li et al., 2023; Javaheripi et al., 2023; Abdin et al., 2024; Maini et al., 2024). Our goal here is not to generate new content but to clean up the existing content through careful prompting. We use three techniques proposed in the literature. All synthetic data was generated using a Mistral-Instruct-7b-v0.1 model with the following sampling parameters:

- Temperature: 0.7
- Top-p (nucleus sampling): 0.95

These parameters were chosen to balance creativity and coherence in the generated text.

We implemented distinct pipelines that represent leading methodologies in synthetic data generation. We modified the prompts from the original work (if available) to promote better format-following and encourage longer, high-quality text. Generation procedures are detailed below with full prompts provided in boxes A.2.2.1-A.2.2.4.

- **High Quality Rephrasing (HQ)** Inspired by WRAP (Maini et al., 2024), we prompt LLM to rewrite source documents into clear, coherent, and well-structured text.

- **Question Answering Rephrasing (QA)** Inspired by WRAP (Maini et al., 2024), we prompt LLM to convert source document into a conversational QA format.

- **Textbook-style Rephrasing (TB)** Inspired by family of Phi models (Li et al., 2023; Javaheripi et al., 2023; Abdin et al., 2024), we first convert text into book chapter titles and then

prompt the LLM to generate new content for each chapter, with variations in prompts for different target audiences (grade school, college, expert, general).

Light heuristic post-filtering was applied to all generated synthetic datar, removing documents that were excessively short (e.g., less than 50 tokens) or excessively long relative to the target length for that generation type, if such outputs occurred despite prompt length guidance. The goal of this light filtering was to remove egregious generation errors without overly sanitizing the data or significantly altering its distribution.

---

### A.2.2.1 Prompt Template HQ Rephrasing

- **System Prompt:** Provide direct and detailed response to the instructions without adding additional notes.
- **[USER]:** For the following document, regardless of its original content or formatting, write a full article of the same content in high quality English language as in texts on Wikipedia: [xxxx]. Provide the rephrased article without any additional notes. Long article with full length and complete details. Rephrased article:

---

### A.2.2.2 Prompt Template QA Rephrasing

- **System Prompt:** Provide direct and detailed response to the instructions without adding additional notes.
- **[USER]:** For the following document, regardless of its original content or formatting, convert it into a comprehensive list of question-answer pairs with multiple tags of "Question:" followed by "Answer:", where questions and answers cover complete information of the original document. Document: [xxxx]. Provide the converted question-answer pairs without any additional notes. Question-answer pairs with corresponding tags ("Question:", "Answer:"):

---

### A.2.2.3 Prompt Template for Generating Textbook-style Synthetic Data: Step 1, Outline Generation

- **Step 1: generate an outline based on input text.**
- **System Prompt:** Provide direct and detailed response to the instructions without adding additional notes.
- **[USER] <4 versions>:** Imagine you are a prolific author tasked with writing a textbook. You are working on writing a textbook involving the knowledge and information of the following text. Text: [xxxx]\n Your task is to write an outline for the textbook. Your target audiences are <grade school students/college students/field experts/general public>. The textbook has 10 chapters in total plus title, introduction, and appendices. Textbook outline:

---

### A.2.2.4 Prompt Template for Generating Textbook-style Synthetic Data: Step 2, Chapter Generation

- **Step 2: generate each section based on outline.**
- **System Prompt:** Provide a direct and detailed response to the instructions without adding additional notes.
- **[USER]:** Imagine you are a prolific author tasked with writing a textbook. You are working on writing a textbook with the following outline.\n Outline: [xxxx] \n Your task is to write Chapter x of the textbook. Your target audiences are grade school students. Include exercises at the end of the chapter to test the reader's knowledge of the chapter and then provide reference answers to each question.

---

## B    EVALUATION DATASET

Unlike prior scaling law works that report training loss (Hestness et al., 2017; Hoffmann et al., 2022) or test loss on a held-out validation set (Kaplan et al., 2020) from training distribution, we measure upstream loss on a held-out test set from original CC as well as a diverse set of 16 non-code/math

Table 3: Model configuration parameters for different scale sizes.

| Model | Hidden Dim | #Layers | #Heads | Batch Size | Grad Acc | DP | TP | #Params |
|-------|-----------|---------|--------|------------|----------|----|----|---------|
| 100m | 576 | 7 | 9 | 4 | 8 | 1 | 1 | 175,628,736 |
| 200m | 832 | 10 | 13 | 4 | 8 | 1 | 1 | 298,632,256 |
| 500m | 1280 | 16 | 20 | 4 | 8 | 1 | 1 | 653,436,160 |
| 1b | 1792 | 22 | 28 | 4 | 8 | 1 | 1 | 1,317,616,384 |
| 2b | 2240 | 28 | 35 | 4 | 8 | 1 | 1 | 2,292,740,800 |
| 3b | 2624 | 32 | 41 | 2 | 8 | 2 | 1 | 3,360,234,048 |
| 4b | 2816 | 34 | 44 | 1 | 8 | 4 | 1 | 4,103,539,968 |
| 6b | 3200 | 40 | 50 | 1 | 1 | 32 | 1 | 5,801,833,600 |
| 8b | 3648 | 45 | 57 | 1 | 1 | 32 | 1 | 8,122,355,904 |
| 11b | 4096 | 51 | 64 | 2 | 1 | 16 | 2 | 11,372,228,608 |

English text domains from The Pile (Gao et al., 2020). Because we use scaling laws for comparative analysis across data interventions, it is critical to assess model performance on external validation sets to enable fair and meaningful comparisons across different training datasets.

## C  MODEL DESIGN

### C.1  LIST OF TRAINED MODELS

We adopt a standard transformer-based model architecture based on LLaMA3 (Grattafiori et al., 2024) for all of our scaling analysis. In Table 3 we list the model size and configuration of all models used in this study.

### C.1.1  TRAINING AND EVALUATION HYPERPARAMETERS

We trained all models from scratch using the *Meta Lingua* library (Videau et al., 2024) across one or multiple nodes depending on model size. We use AdamW (Kingma & Ba, 2014) optimizer with $\beta_1 = 0.9$, $\beta_2 = 0.95$, and a weight decay of 0.1, paired with a cosine schedule and 10% linear warmup. All runs used a 4096-token context length, a 1M-token effective batch size, and the Llama 3 TikToken tokenizer (128k vocab) (Grattafiori et al., 2024). Table 4 and Table 5 list hyperparameters for training and evaluation. Table 3 lists local batch size, gradient accumulation, data parallelism (DP) and tensor parallelism (TP) employed for each model size. These parameters are chosen such that global batch size remains at 1M token across all experiments.

Table 4: Training Hyperparameters

| Hyperparameter | Value |
|----------------|-------|
| Optimizer | AdamW |
| Peak Learning Rate | $3e-4$ |
| Min. LR Ratio | $1e-6$ |
| Warmup Steps | 10% |
| Gradient Clipping | 1.0 |
| Sequence Length | 4096 |
| Effective Batch Size | 1M tokens |
| Prefetch Size | 1024 |
| Add BOS token | True |
| Add EOS token | True |
| Model Data Type | bf16 |
| Epochs | 1 |
| GPU Hardware | NVIDIA A100 80GB |

Table 5: Hyperparameters for Perplexity Evaluation

| Hyperparameter | Value |
| --- | --- |
| Max Tokens to Generate | 1024 |
| Generator Data Type (dtype) | bf16 |

## D    DETAILS ON SCALING ANALYSIS SETUP

Unless otherwise stated, all scaling results reported in the main body of the paper are obtained using a **joint scaling-law fit** of the following parametric form:

$$L(N, D) = A \cdot N^{-\alpha} + B \cdot D^{-\beta} + E.$$

This section describes the methodology used throughout the paper for fitting this form. An exception is Appendix G, where we adopt several engineering practices commonly used in prior Chinchilla-style analyses to improve fitting stability.

We empirically estimate the parameters by fitting this function to the validation loss of over 100 models, ranging from 100M to 8B (3B) parameters and trained on 100M to 40B (200B) tokens for filtering (synthetic) interventions. Each datapoint is visited only once during training, consistent with standard scaling-law methodology. This avoids confounding effects from data repetition and ensures fair comparison across datasets.

**Curve-fitting     and     Initialization:**    We     fit     the     scaling     law     using `scipy.optimize.curve_fit` (Virtanen et al., 2020), specifically the Trust Region Reflective (`trf`) optimizer (Branch et al., 1999), which supports bounded nonlinear least squares. Initial conditions are drawn from prior work (Besiroglu et al., 2024) that challenged assumptions in the original Chinchilla analysis (Hoffmann et al., 2022):

$$[A, B, \alpha, \beta, E] = [482, 2085, 0.3478, 0.3658, 1.8].$$

**Appendix G Variant:**    We additionally adopt several fitting practices inspired by the Chinchilla methodology to improve fitting quality. These include fitting the amplitude parameters $(A, B)$ in log-space, using a Huber loss to reduce sensitivity to heavy-tailed residuals, detecting and removing statistical outliers prior to fitting, optimizing with L-BFGS, and estimating parameter uncertainty via bootstrap resampling (1000 samples).

**Parameter Count Definition:**    There exists inconsistency in prior work regarding whether to include embedding parameters in the total parameter count $N$. Kaplan et al. (2020)'s scaling law analysis excludes embedding parameters, while the Hoffmann et al. (2022)'s analysis includes them. We examined both conventions and found that qualitative trends and conclusions remain consistent. For consistency, we report results using the total parameter count *including* embeddings.

## E    SCALING LAW COMPONENT ANALYSIS

In Section 4, Figure 1, we showed the impact of a handful of data quality interventions on components of scaling law. Here we show the impact of all 23 datasets from QualityPajama benchmark suite. We group the results based on the type of interventions (heuristic filters vs. synthetic). We also compare the best from each group.

### E.1    HEURISTIC FILTERS

To study the effect of heuristic-based data quality interventions on scaling behavior, we apply a sequence of commonly used filters, including NSFW removal, aesthetic filtering, PageRank-based filtering, deduplication at varying similarity thresholds, and grammar-based filtering via average sentence length. These filters are chosen based on their frequent use in high-quality dataset pipelines such as C4 (Raffel et al., 2020), Dolma (Soldaini et al., 2024), and FineWeb (Penedo et al., 2024).

For each filter, we evaluate its impact on the scaling law parameters by comparing the fitted values before and after its application, as well as across different configurations (e.g., similarity thresholds for deduplication or percentile cutoffs for PageRank). Detailed analyses are shown in Figures 7, 8, and 9.

## E.2 SYNTHETIC DATA GENERATION

To evaluate the impact of synthetic data interventions on scaling behavior, we curate datasets using three prompting strategies: high-quality rephrasing (HQ), question-answer transformation (QA), and textbook-style rewriting (TB). These methods draw inspiration from prior work on synthetic pretraining data (Maini et al., 2024; Li et al., 2023; Javaheripi et al., 2023; Abdin et al., 2024), and are applied using the Mistral-Instruct-7B model (Jiang et al., 2023). We mix synthetic data with natural data at different ratios (e.g., 33% synthetic, 67% original). We fit scaling laws on these synthetic variants to analyze how text rewriting influences parameter stability and scaling behavior. Detailed results are shown in Figure 10, 11, 12, and 13.

## F  LIMITATION OF ZIPFIAN DISTRIBUTION THEORY

While prior work has suggested that token distribution characteristics—such as Zipfian structure could explain power law exponent's behavior, our empirical findings show that this theory may not be sufficient to explain the variation in power-law exponents. We analyzed token frequency distributions across our filtered datasets (Figure 14) and found that the Zipfian exponents are weakly negatively correlated with the model size exponent $\alpha$ (correlation = $-0.37$), and show little to no correlation with the data size exponent $\beta$ (correlation = $-0.005$). Table 6 shows scaling exponents across datasets. In several cases, datasets with nearly identical token distributions exhibit substantially different scaling behavior. This suggests that simple distributional statistics, such as Zipfian exponents, fail to capture the deeper structural or semantic properties that influence scaling dynamics. It is possible that higher-order $n$-gram patterns or conceptual structures provide a more explanatory signal.

| Filter | Zipf Exponent ($z$) | Scaling Exponent ($\alpha$) | Scaling Exponent ($\beta$) |
|---|---|---|---|
| high_perplexity | 1.1820 | 0.3509 | 0.2536 |
| nsfw | 1.0950 | 0.4341 | 0.1982 |
| aesthetic | 1.0907 | 0.4097 | 0.1946 |
| deduped_0.7 | 1.3898 | 0.3633 | 0.2173 |
| deduped_0.9 | 1.3938 | 0.3392 | 0.2393 |
| deduped_1.0 | 1.1638 | 0.3499 | 0.2093 |
| no_pr | 1.0599 | 0.3884 | 0.1855 |
| low_pr | 1.3943 | 0.3772 | 0.1805 |
| med_pr | 1.3744 | 0.4157 | 0.1982 |
| high_pr | 1.2625 | 0.3499 | 0.1826 |
| medium_text | 1.3541 | 0.3740 | 0.1885 |
| long_text | 1.2944 | 0.4106 | 0.1833 |

Table 6: Zipf exponent $z$ and scaling law exponents $\alpha$ and $\beta$ across different data filters.

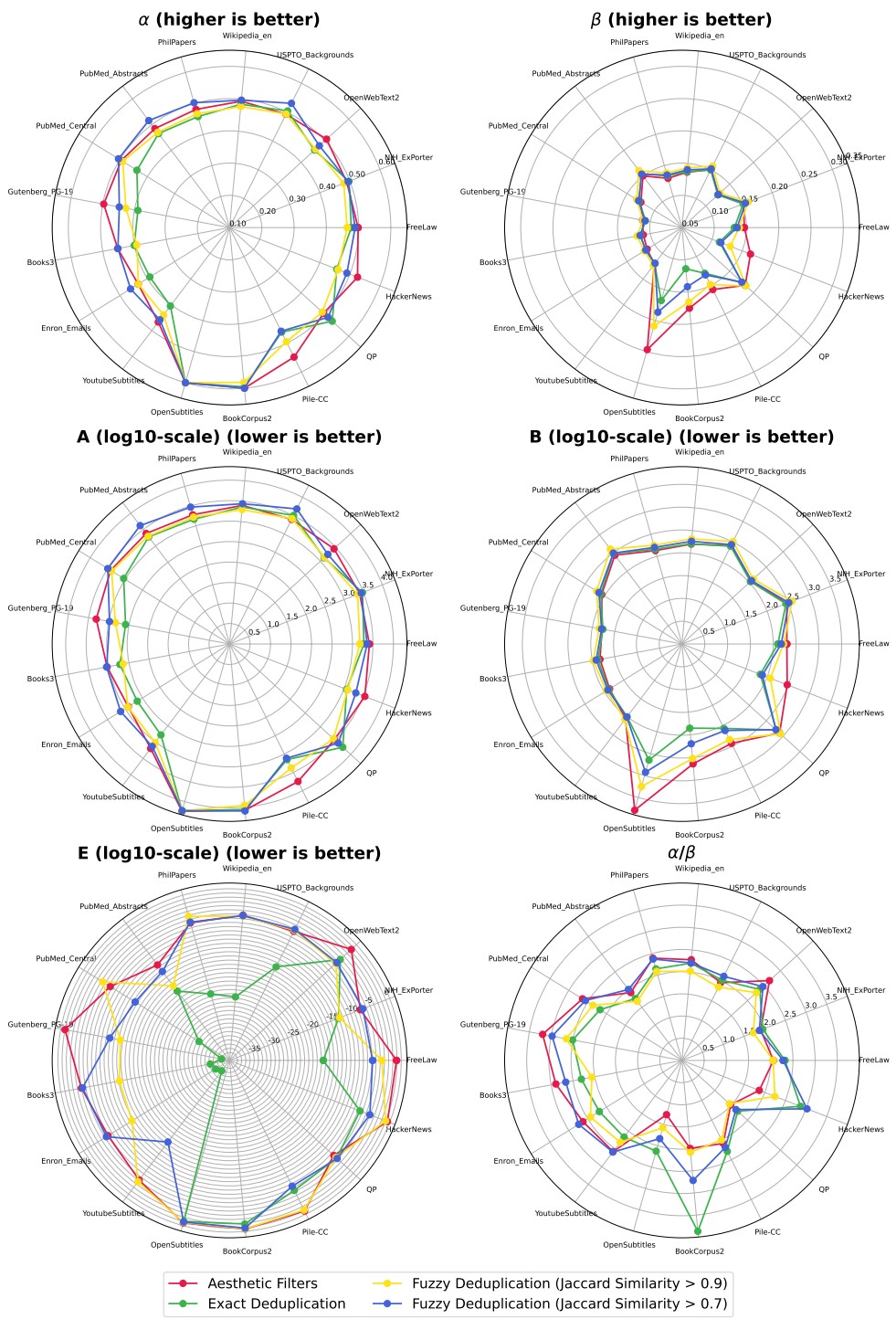

Figure 7: **How does deduplication affect scaling law components?** The red line marks the dataset before any deduplication is applied. Other lines represent deduplicated variants using different similarity thresholds.

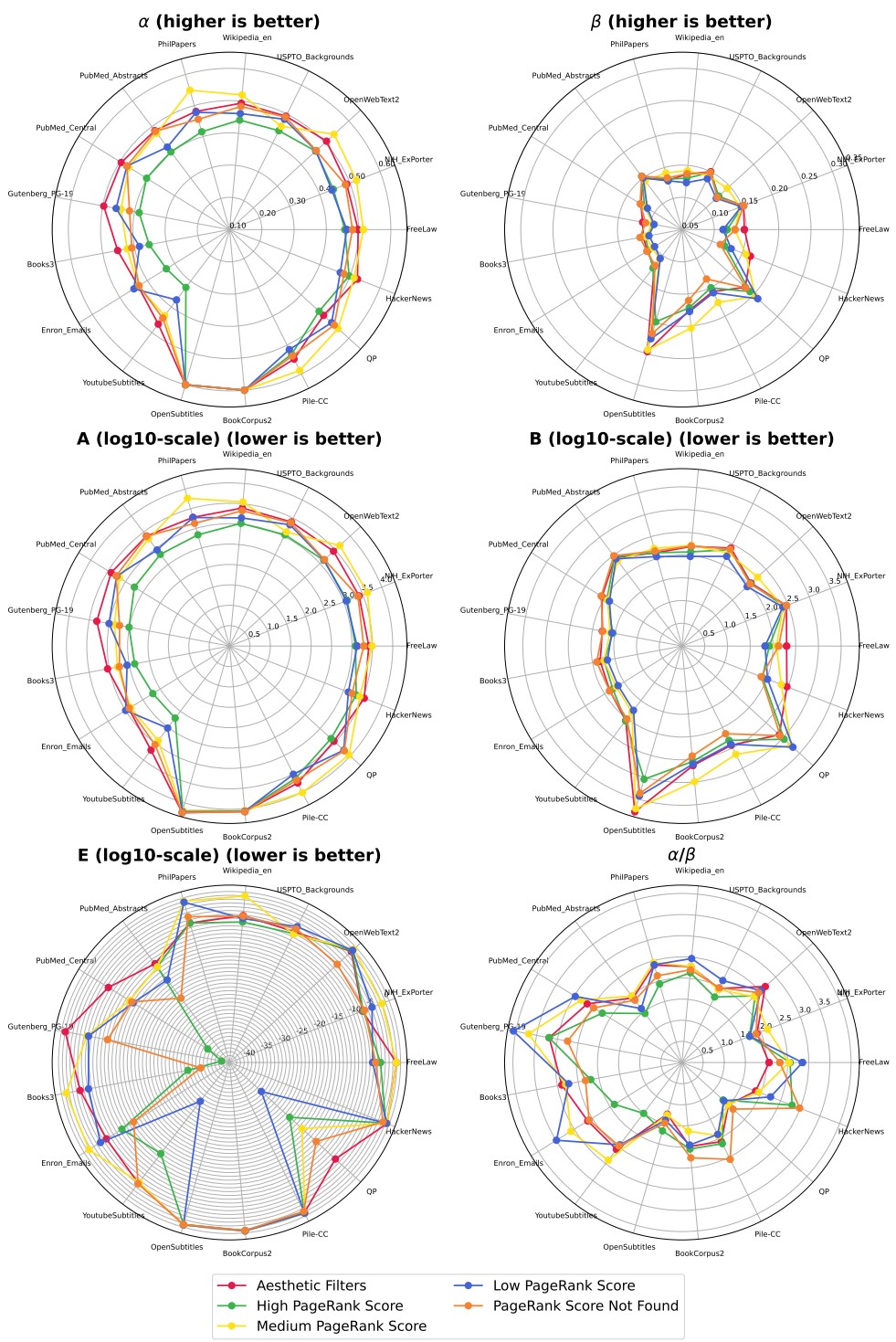

Figure 8: **How does PageRank-based filtering affect scaling law components?** The red line denotes the dataset before applying any PageRank filters. Other lines correspond to thresholds applied to the PageRank score. `Low PageRank` retains pages with scores below $X$, `High PageRank` retains those above $Y$, and `Medium PageRank` keeps pages between $X$ and $Y$. `PageRank Not Found` includes pages missing from the reference PageRank table. Thresholds $X$ and $Y$ are set to the $33^{rd}$ and $67^{th}$ percentiles of the score distribution of pages in the PageRank table.

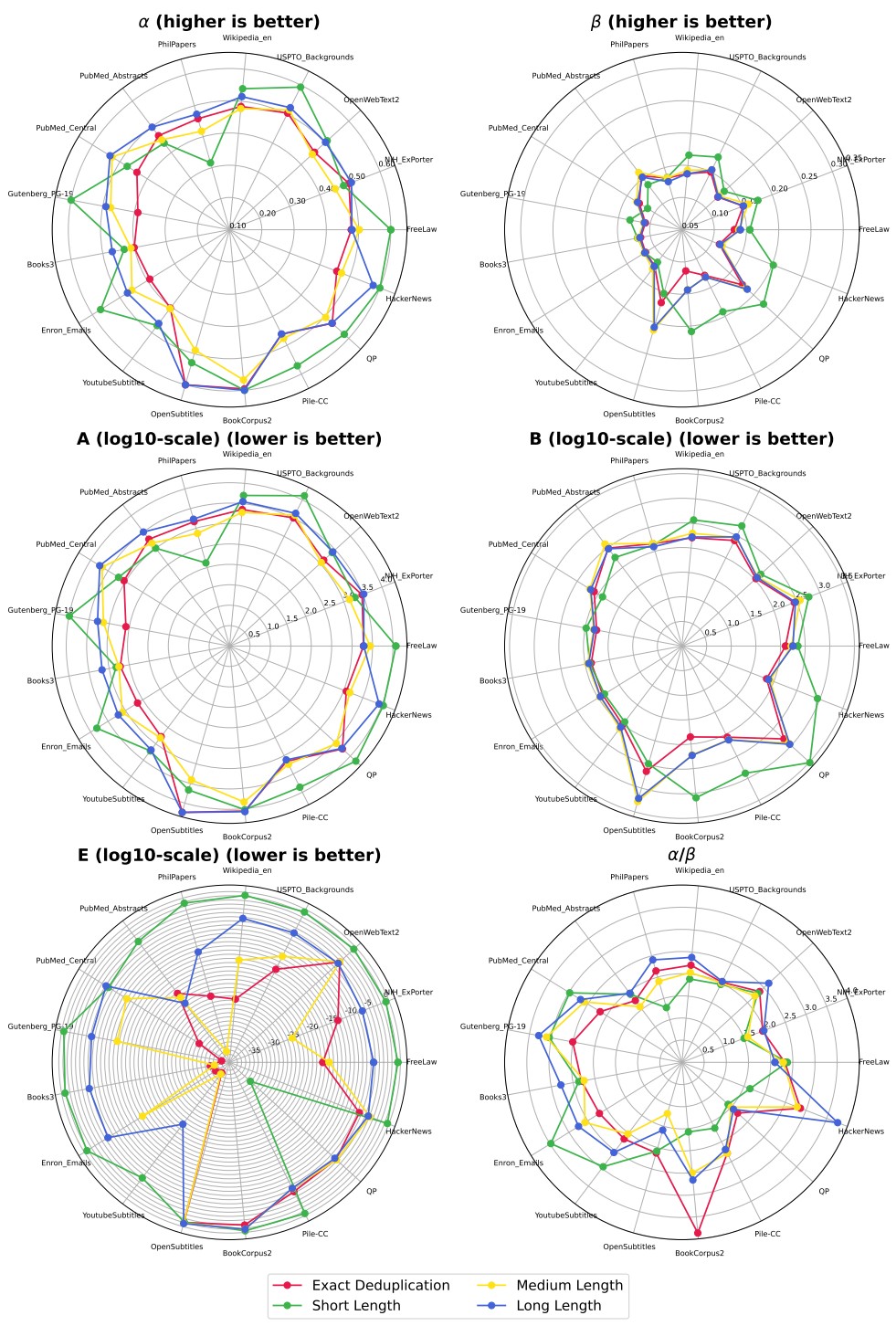

Figure 9: **How does grammar complexity (average sentence length) affect scaling law components?** The red line indicates the dataset before applying any sentence length filters. Datasets are filtered based on average sentence length, with thresholds set at 10 tokens for `short text` and 25 tokens for `medium text`.

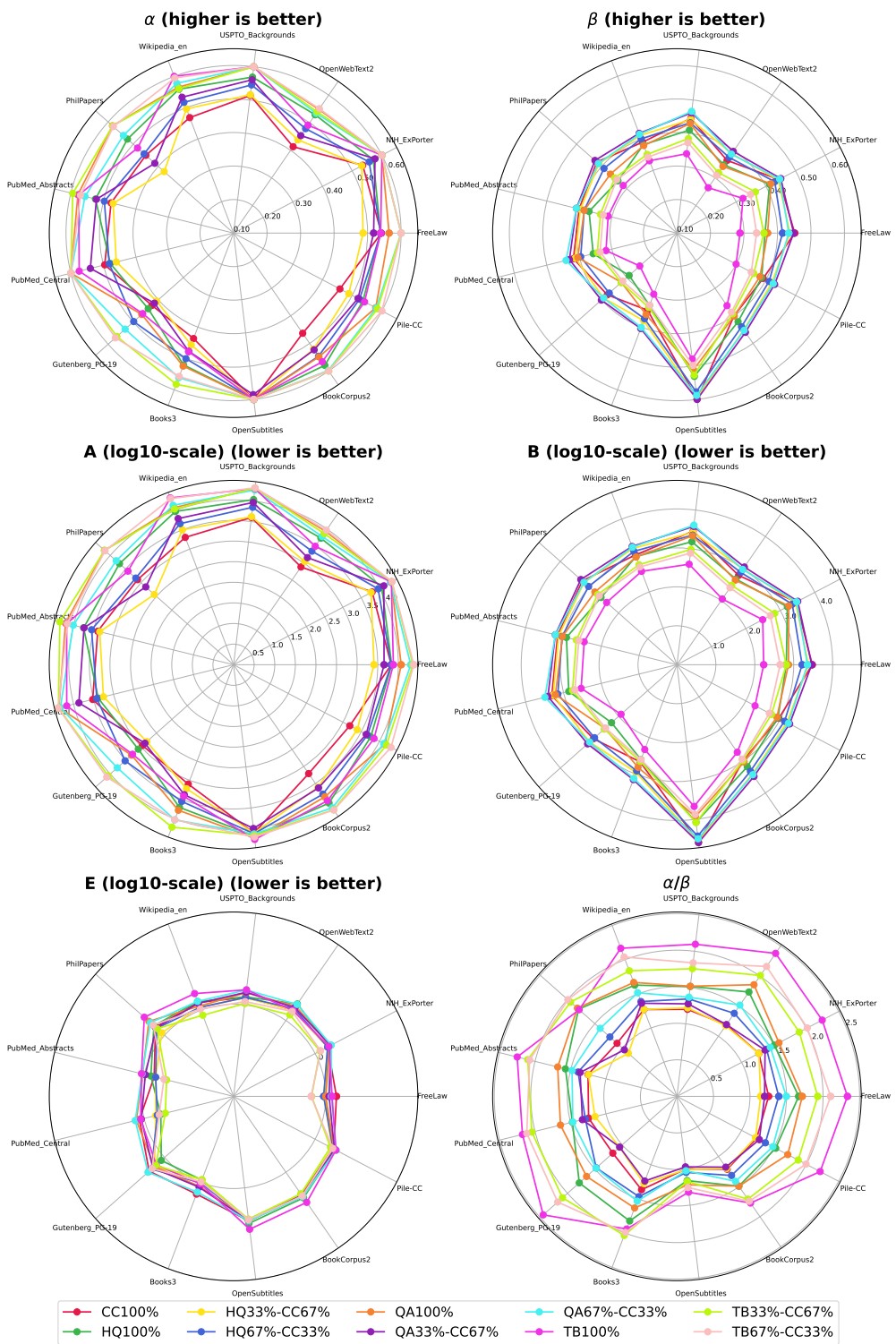

Figure 10: **How does synthetic data influence scaling law components?** Different lines show different synthetic data generation techniques and mixing ratio, and along the radial axis we have the validation set.

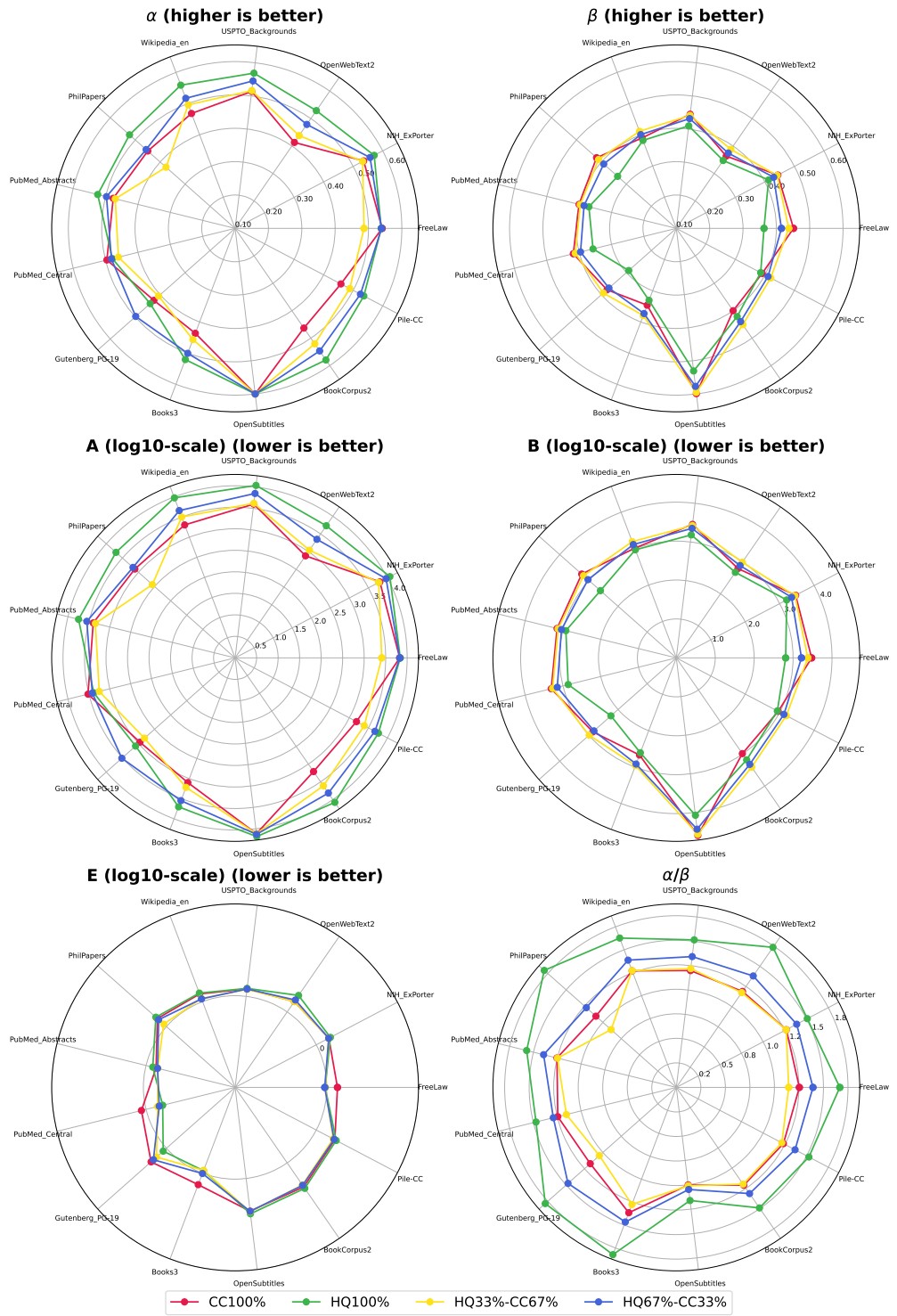

Figure 11: **How does HQ synthetic data generation influence data quality?** HQ refers to high-quality rephrasing, and CC refers to the raw natural Common Crawl dataset. HQ[N]-CC[M] refers to the mixture of synthetic and natural and N and M captures the percentage.

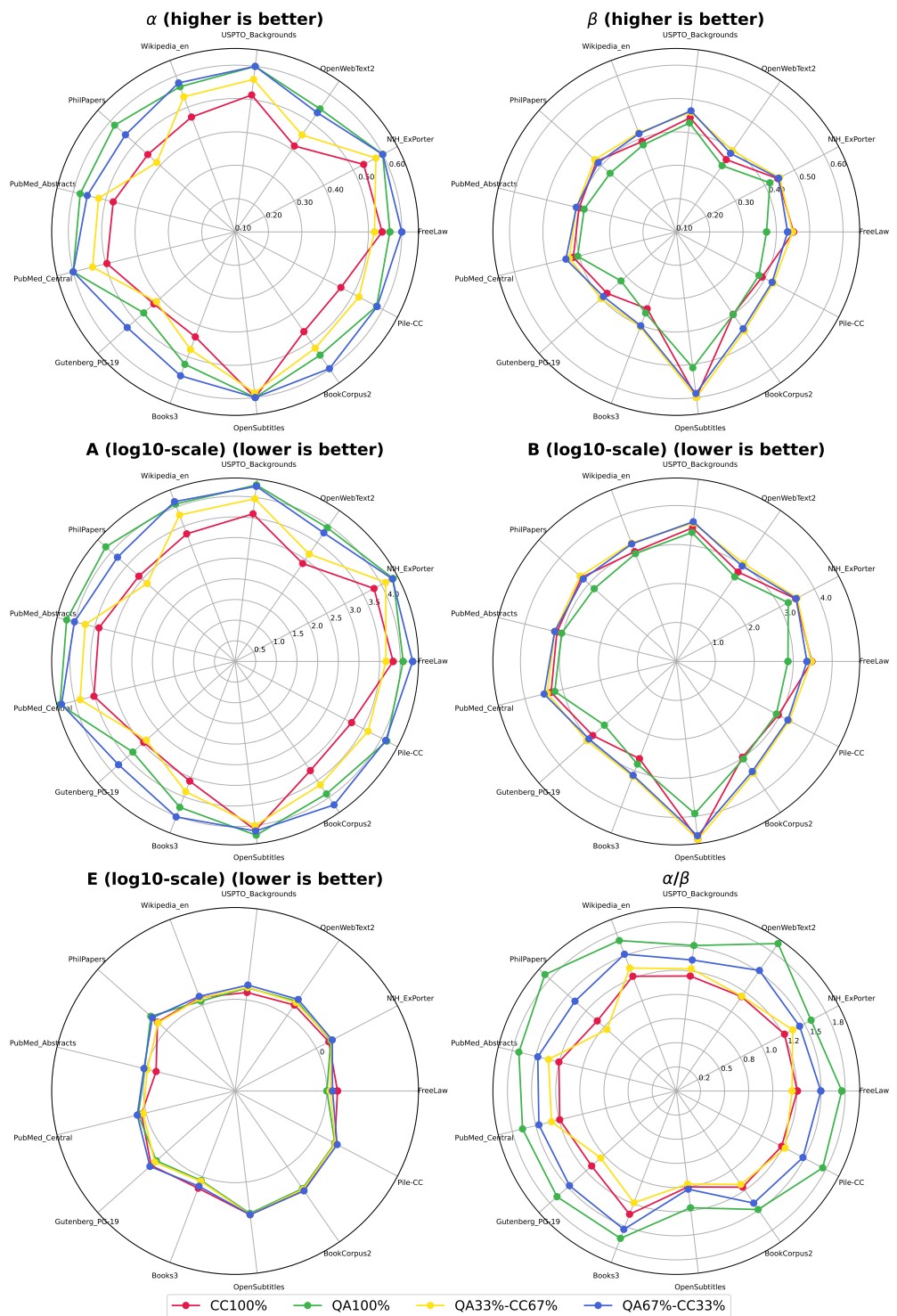

Figure 12: **How does QA synthetic data generation influence data quality?** QA refers to Question-Answering rephrasing, and CC refers to the raw natural Common Crawl dataset. QA[N]-CC[M] refers to the mixture of synthetic and natural and N and M captures the percentage.

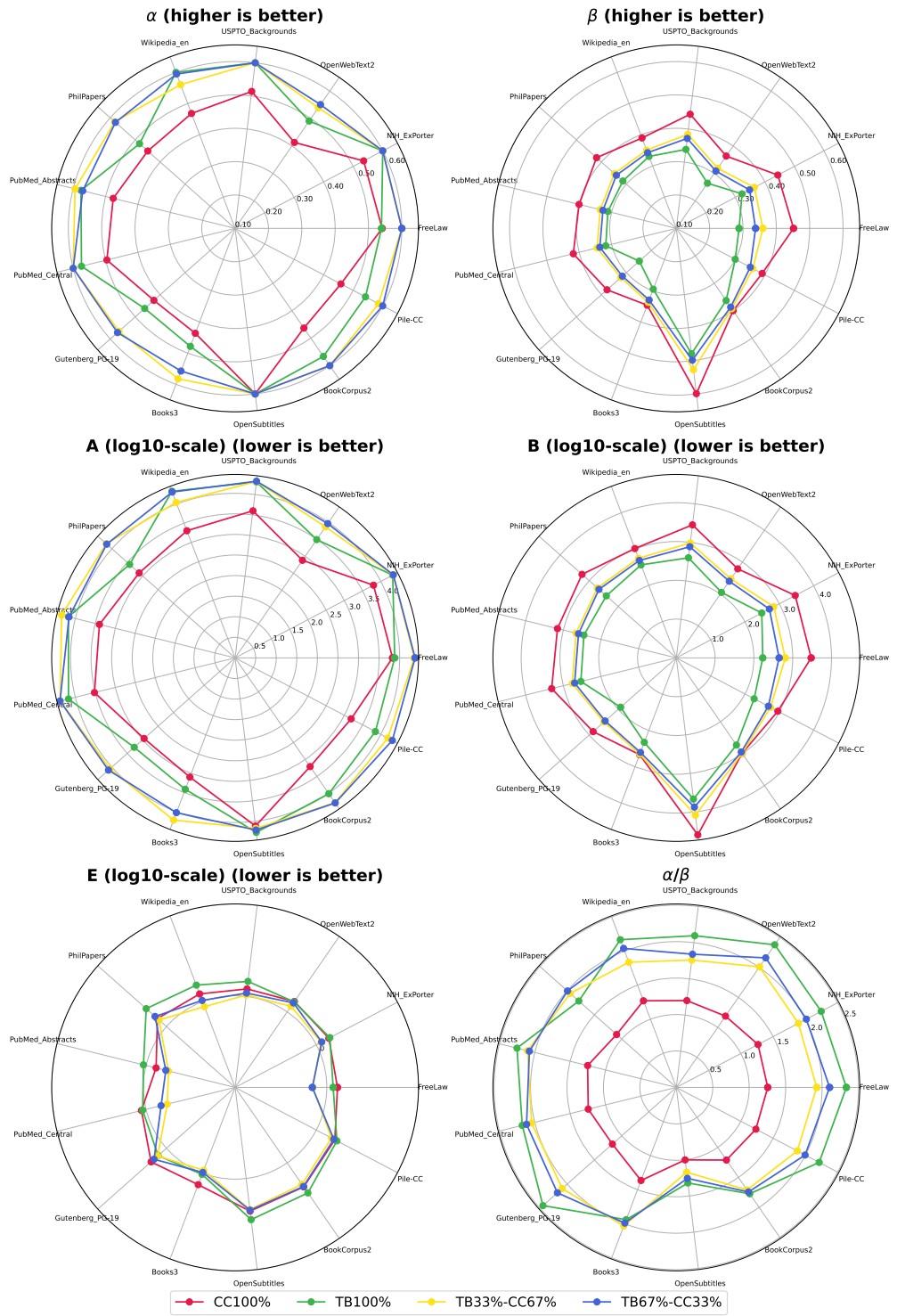

Figure 13: **How does TB synthetic data generation influence data quality?** TB refers to Textbook-style rephrasing, and CC refers to the raw natural Common Crawl dataset. TB[N]-CC[M] refers to the mixture of synthetic and natural and N and M captures the percentage.

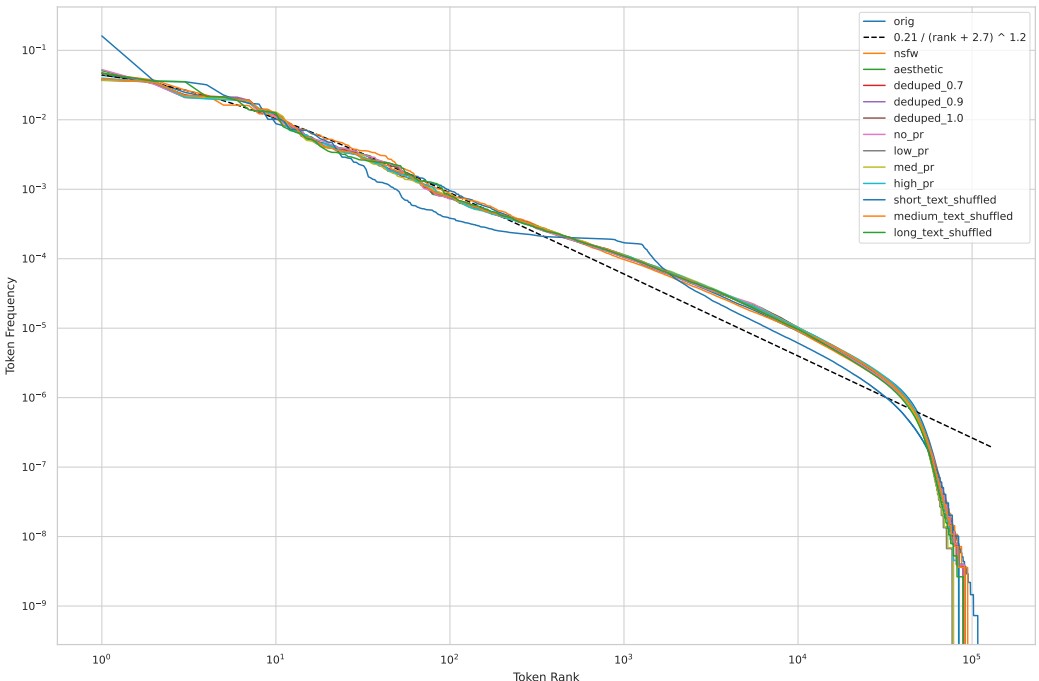

Figure 14: Token distribution across different QualityPajama datasets

## G    BEYOND CHINCHILLA: STABLE SCALING-LAW FITS UNDER TRAIN–TEST DRIFT

We include this appendix to disentangle how much of the observed variation in scaling-law parameters across datasets reflects true distributional differences versus fit instability of the scaling-law form itself. This investigation revealed that the standard Chinchilla scaling-law form fails to predict performance reliably when train and test distributions differ. It also exhibits severe parameter instability (large coefficients of variation, high CV) when the dataset lacks sufficient coverage in the large-$N$/large-$D$ regime. We propose a new scaling-law formulation that remains identifiable with fewer datapoints, avoiding the high cost of collecting expensive large-scale measurements. Empirically, this formulation is substantially more robust than the traditional Chinchilla form under distribution shift and limited-data settings. A key ingredient for robust estimation is incorporating the theoretical asymptotic loss floor. Finally we show that while the standard Chinchilla form exhibits weak identifiability under distribution shift, stabilized fits recover the same qualitative scaling trends across data interventions, suggesting that the exponent and coefficient shifts reported in the main paper reflect true data distribution effects rather than fitting artifacts (Figure 24).

### G.1    INTRODUCTION

Classical neural scaling-law formulations implicitly assume that training and evaluation data are drawn from the same underlying distribution. In practice, scaling laws are often used when train and test distributions differ—sometimes substantially. As data curation pipelines evolve or filtering strategies become more aggressive, the training distribution can drift away from the target evaluation distributions for which we ultimately want predictions. This mismatch violates the assumptions of standard scaling-law forms and directly impacts parameter identifiability.

A second pain point arises from the need for high-compute datapoints to accurately estimate the irreducible loss floor $E$. When the data do not extend far enough into the large-$N$/large-$D$ regime, the asymptotic term becomes weakly identified (Fig. 15)—and the situation worsens when train/test distributions drift apart (Fig. 16).

In this appendix, we document a step-by-step journey toward a more robust scaling-law form under limited datapoints: we begin by diagnosing failure modes in the baseline Chinchilla form and then describe the empirical insights that guided subsequent refinements.

## G.2 METHODOLOGY

**Dataset.** We evaluate scaling-law forms across 299 train–test pairs, formed by the Cartesian product of 23 training sets from *QualityPajama* and 13 test sets (12 from the Pile and 1 validation set drawn from the training distribution). For each train–test pair, we have $\sim 40$ datapoints with approximately $9 \times 14$ $N \times D$ coverage, with model sizes ranging from 20M to 3B parameters ($\sim 2$ orders of magnitude) and data sizes from 100M to 38B tokens ($\sim 2.6$ orders of magnitude).

**Measure of success.** We define a composite quality score (0–100 points) to compare scaling-law models, balancing prediction stability and predictive performance while penalizing known pathological solutions. We weight prediction stability (40 points) and prediction quality (40 points total) equally, reflecting the requirement that a useful scaling law must both fit the data well and yield stable predictions. The remaining 20 points penalize degenerate asymptotic behavior. The score aggregates:

1. **Performance Stability (40)**: fraction of train–validation pairs where the *parametric* prediction CV (via Monte Carlo propagation of parameter uncertainty) is below 1.0.

2. $E$**-Collapse Avoidance (20)**: rewards models with irreducible error satisfying $E \geq 0.1$, penalizing pathological $E \to 0$ solutions.

3. $R^2$ **Performance (20)**: out-of-bag predictive accuracy on a linear scale, with $R^2 = 1$ receiving full credit.

4. **RMSE Performance (20)**: absolute prediction error normalized between best and worst RMSE across all models.

This rubric ensures scaling-law models achieve stable prediction, avoid degenerate asymptotes, and maintain strong predictive accuracy; underperformance on any dimension substantially reduces the overall score.

## G.3 COVERAGE–IDENTIFIABILITY DIAGNOSTICS IN THE CHINCHILLA FORM

**In-distribution evaluation.** Fig. 15 shows parameter uncertainty as a function of data coverage for the Chinchilla form under in-distribution evaluation (orig→val). Each heatmap reports the coefficient of variation $\mathrm{CV} = \mathrm{std}/\mathrm{mean}$ for parameters when fitting subsets restricted to $N \leq N_{\max}$ and $D \leq D_{\max}$, computed from 150 bootstrap samples. Dark regions correspond to stable, well-constrained estimates; bright regions indicate high uncertainty ($\mathrm{CV} < 0.1$: excellent; 0.1–0.2: acceptable; $> 0.2$: poor). Notably, the asymptotic term $E$ exhibits a sharp identifiability transition: it collapses to zero when $N_{\max} < 2\mathrm{B}$ or $D_{\max} < 5\mathrm{B}$, and becomes non-zero and identifiable only with sufficient large-$N$/large-$D$ coverage. In contrast, $(A, B, \alpha, \beta)$ are comparatively robust, with uncertainty decreasing smoothly as coverage increases.

**Distribution shift.** Fig. 16 repeats the analysis under distribution shift (aesthetic→math). Cross-domain evaluation induces severe parameter instability relative to in-distribution evaluation. The asymptotic term $E$ collapses to zero across all coverage levels. Other parameters also degrade sharply: $\alpha$ is unstable in 67/100 cells, $B$ is highly unstable throughout, and $A$ shows extreme variability even in the high-$N$/high-$D$ region; only $\beta$ retains moderate stability. Together, these results indicate that distribution shift fundamentally destabilizes scaling-law parameters: even with large models and extensive training data, parameters learned in one domain do not reliably predict loss in out-of-distribution settings.

## G.4 MOTIVATION: ASYMPTOTIC FLOOR COLLAPSE IN THE CHINCHILLA FORM

Using the standard 5-parameter Chinchilla model, we observe that $59\%$ of train–validation pairs produce near-zero asymptotes ($E < 0.1$; Fig. 18). This behavior is incompatible with theory:

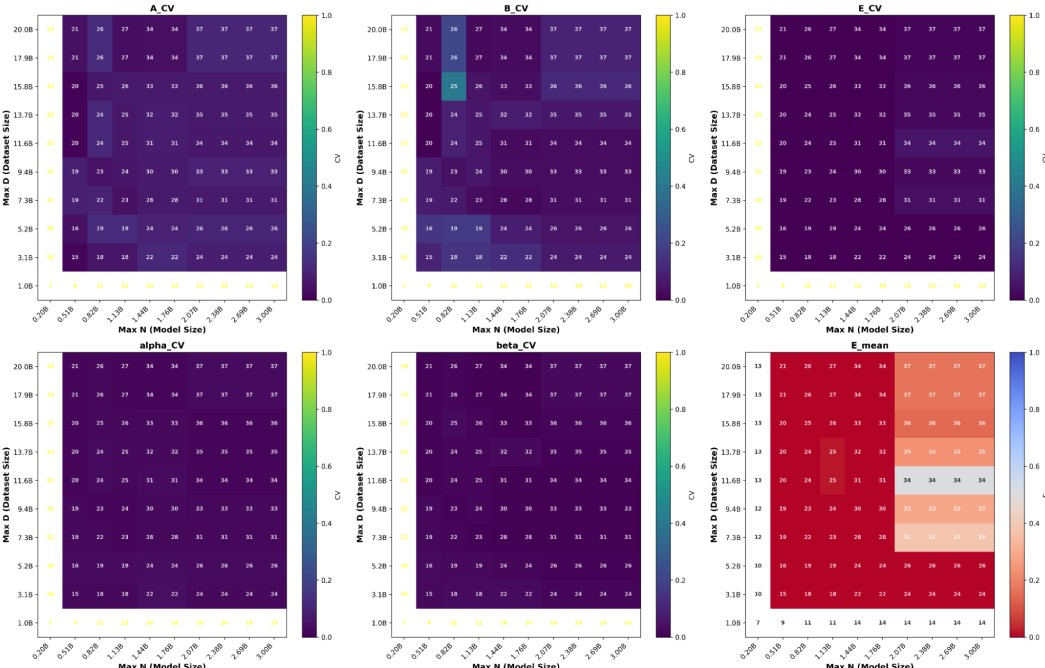

Figure 15: **Parameter uncertainty vs. data coverage in the Chinchilla form (in-distribution; orig→val).** Each heatmap shows coefficient of variation (CV $=\mathrm{std}/\mathrm{mean}$) of parameter estimates when fitting the Chinchilla form to subsets restricted to $N \leq N_{\max}$ and $D \leq D_{\max}$. Panels report CVs for $A, B, \alpha, \beta, E$ from 150 bootstrap samples; the bottom-right panel reports the mean estimated $E$. Numbers indicate datapoints used. Dark regions indicate stable estimates; bright regions indicate high uncertainty (CV $< 0.1$: excellent; 0.1–0.2: acceptable; $> 0.2$: poor). $E$ exhibits a sharp identifiability transition, collapsing to zero when $N_{\max} < 2\mathrm{B}$ or $D_{\max} < 5\mathrm{B}$ and becoming identifiable only with sufficient large-$N$/large-$D$ coverage.

when train and test distributions diverge, we expect the asymptotic floor to be strictly positive and approximated by

$$E \approx H_{\mathrm{val}} + \mathrm{KL}(p_{\mathrm{val}} \,\|\, p_{\mathrm{train}}). \tag{1}$$

The collapse indicates that, without additional information, the model prefers to "explain away" the asymptote by reallocating mass into $A$ and $B$. The failure is most severe when (i) train and validation distributions diverge (77% failure for high JS-divergence pairs vs. 56% for low JS-divergence pairs) and/or (ii) we lack high-compute datapoints to anchor the asymptote (82% failure with $< 30$ points vs. 26% with $> 45$ points).

While JS divergence computed from unigram distributions is only a rough proxy for train–test drift, $E$-collapse rates vary sharply across validation domains. Technical and specialized domains (code, foreign languages, academic papers) exhibit $E$-failure rates exceeding 90%, likely reflecting limited representation in the Common Crawl-based training corpus (Fig. 17).

### G.5 PROGRESSIVE IMPROVEMENTS VIA ITERATIVE REFINEMENT

Fig. 18 summarizes iterative improvements in robustness. We report out-of-bag $R^2$ and RMSE (top), catastrophic failure rates for prediction CV ($> 1$) and for $E$ ($< 0.1$) (bottom-left), and the aggregate quality score (bottom-right). Starting from the baseline 5-parameter Chinchilla form, we add $E$-regularization around a theoretical bound, and finally consider a 7-parameter extension with scale-dependent amplitude terms. Both the 5-parameter $E$-regularized model and the 7-parameter model substantially outperform the baseline Chinchilla formulation (71.17 and 71.84 vs. 60.40 points). Given the negligible difference (0.67 points), we select the 5-parameter $E$-regularized model as the preferred model by Occam's razor: comparable predictive quality with fewer parameters and improved interpretability.

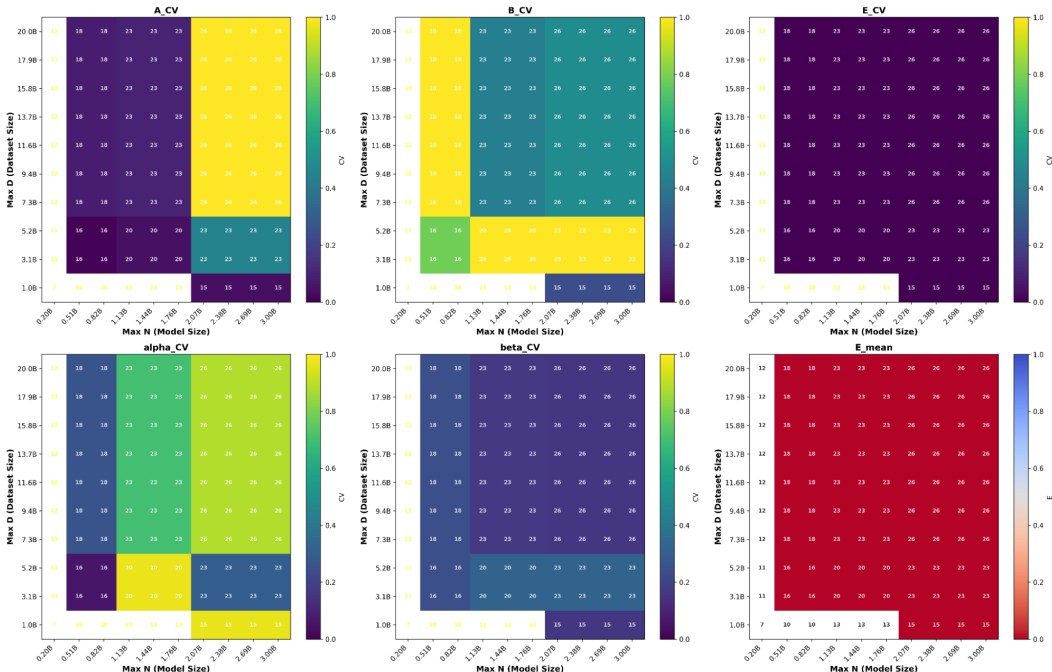

Figure 16: **Parameter uncertainty vs. data coverage in the Chinchilla form (distribution shift; aesthetic→math).** Cross-domain evaluation induces severe parameter instability: $E$ collapses to zero across all coverage levels; $\alpha$ and $B$ become highly unstable, and $A$ shows extreme variability even at large $N$ and $D$. Only $\beta$ remains moderately stable.

### G.6 DISTANCE-DEPENDENT $E$ REGULARIZATION

To address $E$-collapse in the baseline model (59.2% of pairs with $E \to 0$), we introduce an information-theoretic regularization strategy that (i) pulls $E$ toward a theoretical floor and (ii) adapts regularization strength based on train–validation distribution distance.

**Theoretical prior.** We set

$$E_{\mathrm{prior}} = 0.05 \cdot (H_{\mathrm{val}} + \mathrm{KL}_{\mathrm{div}}) \cdot \ln(2), \tag{2}$$

where $H_{\mathrm{val}}$ is Shannon entropy of the validation set unigram distribution and $\mathrm{KL}_{\mathrm{div}}$ is the KL divergence from validation to training unigram distributions. The factor 0.05 is an empirical scaling constant that calibrates information-theoretic quantities to the numerical range of observed neural losses: unigram cross-entropy is an upper bound on irreducible error, while neural models leverage richer contextual structure.

**Regularization strength.** We add a quadratic penalty to the Huber loss objective:

$$R_E = \lambda_E \cdot (E - E_{\mathrm{prior}})^2, \tag{3}$$

with distance-dependent strength

$$\lambda_E = 0.02 + 1.05 \cdot (1 - s)^2, \qquad s = \frac{1}{1 + \mathrm{JS}_{\mathrm{div}}}. \tag{4}$$

This design applies minimal regularization when train and validation are similar ($s \approx 1$ implies $\lambda_E \approx 0.02$), while imposing stronger regularization for distant pairs (e.g., $s \approx 0.6$–$0.7$ yields $\lambda_E \approx 0.12$–$0.17$), preventing collapse.

**Effect.** This approach reduces $E$-collapse from 59.2% to 3.3%, while maintaining excellent fit quality (mean out-of-bag $R^2 > 0.9$). For brevity, we report only the final winning approach; we explored ablations including hard constraints on $E$, direct penalties for small $E$, regularization around alternative targets (e.g. MCMC-estimated $E$), and alternative distance-to-$\lambda_E$ mappings (fixed, linear).

**E-Collapse Failure Rate by Validation Set**
**(Original 5-Parameter Model)**

Figure 17: $E$-**collapse failure rates vary by validation domain.** The baseline 5-parameter model exhibits near-total $E$-collapse (92–100%) for technical/specialized validation sets (code, foreign language, academic papers) but maintains stability (0–23% failure) for general-text domains (movies, books, web crawls). Color coding: red $\geq 70\%$, orange 50–70%, green $\leq 50\%$. Overall: 59% failure across 299 filter–validation pairs.

**Winning recipe (summary).**

$$R_E = \lambda_E \cdot (E - E_{\text{prior}})^2, \tag{5}$$

$$\lambda_E = 0.02 + 1.05 \cdot (1-s)^2, \quad s = \frac{1}{1 + \text{JS}_{\text{div}}}, \tag{6}$$

$$E_{\text{prior}} = 0.05 \cdot (H_{\text{val}} + \text{KL}_{\text{div}}) \cdot \ln(2). \tag{7}$$

### G.7 CHARACTERIZING PREDICTION UNCERTAINTY

The usefulness of scaling laws as forecasting tools depends on prediction uncertainty in unobserved regimes. We distinguish two sources:

**(i) Functional-form uncertainty via Gaussian process diagnostics.** We use a Gaussian process (GP) as a diagnostic for functional misspecification. We treat the scaling law as a parametric mean function and fit a GP to residuals in log–log space. The GP uses a Matérn-$5/2$ kernel over the 2D input $(\log N, \log D)$ with separate length scales, learned via maximum likelihood. Large or structured GP residual uncertainty indicates misspecification; conversely, negligible GP uncertainty (CV $< 1\%$) suggests the parametric form is sufficient and remaining uncertainty arises primarily from parameter estimation.

Figure 18: **Progressive improvements in scaling-law robustness.** (Top-left) out-of-bag $R^2$ and (top-right) out-of-bag RMSE quantify predictive accuracy on held-out data. (Bottom-left) catastrophic failure rates: for predictive performance, failure is the fraction of train–test pairs with prediction $CV > 1$; for $E$, failure is the fraction with $E < 0.1$. (Bottom-right) aggregate quality score combining prediction accuracy, prediction stability, and $E$-collapse avoidance. The baseline 5-parameter model improves substantially with distance-dependent $E$-regularization; a 7-parameter model with scale-dependent amplitudes yields similar overall quality. We select the 5-parameter $E$-regularized model by Occam's razor.

**(ii) Parametric uncertainty.** We estimate parameter uncertainty via bootstrap resampling (1000 samples) and propagate it into predicted loss via Monte Carlo sampling (1000 draws from the bootstrap parameter distribution).

Fig. 19 compares baseline vs. $E$-regularized 5-parameter models across all 299 pairs. $E$-regularization stabilizes amplitudes (especially $A$) while keeping exponent uncertainty low; it reduces median parametric prediction CV ($20\% \rightarrow 16\%$), maintains negligible GP CV ($< 1\%$), eliminates $E$-collapse ($59\% \rightarrow 3.3\%$), reduces catastrophic prediction failures, and improves out-of-bag $R^2$ for 98% of pairs.

## G.8 EXPLORING ALTERNATIVE FITTING FUNCTIONS (WHAT DID NOT WORK)

This section summarizes alternative scaling-law forms explored and abandoned because they offered comparable stability with greater complexity than the 5-parameter model with $E$-regularization.

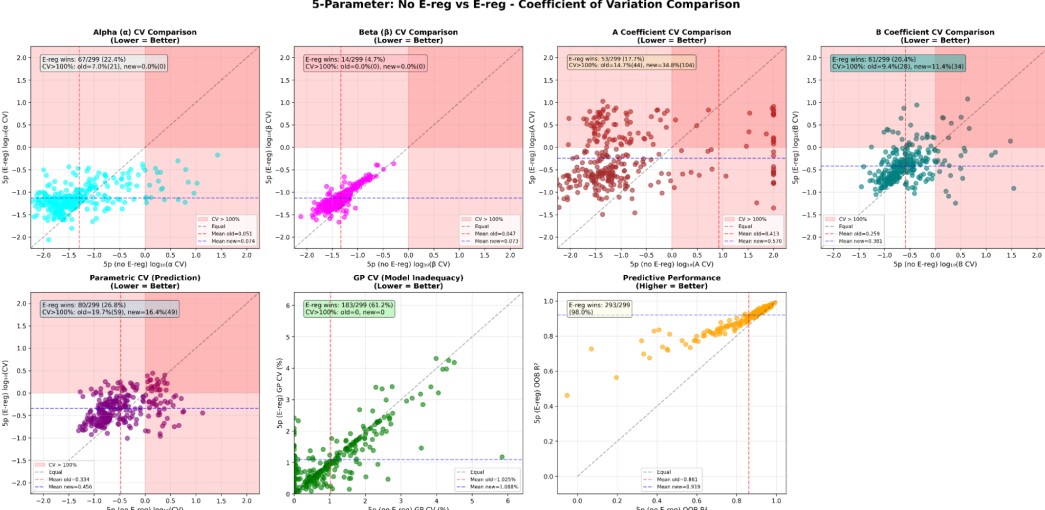

Figure 19: **Prediction uncertainty breakdown: baseline 5-parameter vs. 5-parameter with $E$-regularization.** Top: parameter-level CV for $\alpha, \beta, A, B$. Bottom: prediction CV from parameter uncertainty, GP CV from residual functional uncertainty, and out-of-bag $R^2$. Red shading indicates catastrophic failure (CV > 100%). Dashed lines indicate means/medians.

### G.8.1 BAYESIAN MCMC INFERENCE (NUTS)

We implemented Bayesian MCMC (NUTS) for the 5-parameter Chinchilla form

$$L(N, D) = A\,N^{-\alpha} + B\,D^{-\beta} + E, \tag{8}$$

using log-space transformations for amplitudes to enforce positivity:

$$\log A \sim \mathcal{N}(5, 3), \quad \log B \sim \mathcal{N}(5, 3), \quad \log E \sim \mathcal{N}(0, 2),$$

and truncated normal priors for exponents:

$$\alpha, \beta \sim \text{TruncNormal}(0.25, 0.2;\ 0.01, 0.99).$$

We used a Student-$t$ likelihood ($\nu = 4$) on log-losses to handle heavy tails, ran 4 chains (1000 warmup + 2000 samples, max tree depth 10), and monitored $\hat{R}$ and ESS. We tested both free-$E$ and fixed-$E$ (set to 95% of the minimum observed loss) variants to mitigate the known $A$–$B$–$E$ degeneracy. While MCMC eliminated $E$-collapse, it catastrophically failed in predictive performance: mean out-of-bag $R^2 = -26.42$ across 299 pairs (quality score $-482/100$) and produced highly unstable amplitudes:

- 98% of pairs: catastrophic $A$ failures (CV($A$) > 1),
- 41%: catastrophic $B$ failures (CV($B$) > 1),
- 30%: catastrophic $E$ failures (CV($E$) > 1).

### G.8.2 CROSS-TERM MODULATION VIA SCALE-DEPENDENT AMPLITUDES

Motivated by 1D sliced fits within each train–validation pair—fitting $L(N)$ at fixed $D$ and $L(D)$ at fixed $N$—we observed systematic drift suggesting the Chinchilla form is too rigid over the $(N, D)$ plane:

- $A$ and $\alpha$ vary consistently with $D$ ($R^2 \approx 0.5$),
- $B$ and $\beta$ vary consistently with $N$ ($R^2 \approx 0.3$).

We explored extensions with fully scale-dependent parameters (9+ parameters) but found them unstable given $\sim 30$–50 points per pair. In contrast, allowing only amplitudes to be scale-dependent was supported by the data. The resulting 7-parameter model is

$$L(N, D) = A_0\,D^{\gamma_1}\,N^{-\alpha}\ +\ B_0\,N^{\gamma_2}\,D^{-\beta}\ +\ E, \tag{9}$$

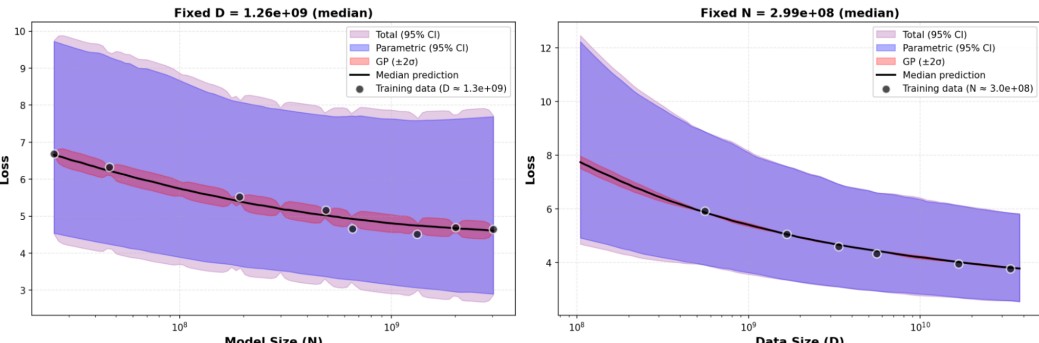

Figure 20: **Prediction uncertainty breakdown for the 7-parameter model (example pair).** Left: fix $D$ at its median and vary $N$. Right: fix $N$ at its median and vary $D$. Black lines show mean predictions from bootstrap-averaged parameters. Shading shows 95% intervals from parametric uncertainty (Monte Carlo over bootstrap parameters) and GP residual uncertainty (Matérn-$5/2$).

with

$$A(D) = A_0 D^{\gamma_1}, \qquad B(N) = B_0 N^{\gamma_2},$$

and $\alpha, \beta$ constant within each pair.

Across 299 pairs, the fitted $\gamma_1$ exhibits a striking domain-specific pattern: pairs with specialized validation sets (math, code, foreign languages) yield $\gamma_1 > 0$, while general-text validation sets yield $\gamma_1 < 0$. Interpreting the sign flip: $\gamma_1 > 0$ implies $A(D)$ increases with data, suggesting that scaling dissimilar data can worsen loss under strong mismatch; $\gamma_1 < 0$ implies $A(D)$ decreases with data, suggesting additional data helps when domains align. This separation helps explain why constant-amplitude Chinchilla fits can fail under distribution shift and motivates cross-term modulation as a mechanism to capture train–test drift.

**Uncertainty breakdown (7-parameter model).** Fig. 20 shows a representative decomposition for a high-similarity pair: GP residual uncertainty remains small and unstructured ($\sigma_{\mathrm{GP}} \sim 0.1$), while parametric uncertainty dominates ($\sigma_{\mathrm{param}} \sim 3$–$4$, $\sim 45\times$ larger). This indicates that instability arises from weak parameter identification rather than functional misspecification. While the 7-parameter model can improve stability for $A$ and $B$, it often shifts instability into the modulation exponents (notably $\gamma_1$), which can degrade overall prediction uncertainty.

### G.9 COVERAGE–STABILITY ANALYSIS OF THE 7-PARAMETER MODEL

We repeat the coverage analysis for the 7-parameter model, analogous to the baseline Chinchilla evaluation. Under in-distribution evaluation (orig→val), the 7-parameter model shows similarly strong stability across parameters (Fig. 21). Under distribution shift (aesthetic→math), it maintains good stability for most parameters, with notable variability in $A_0$, but it recovers stable, non-zero $E$ values (Fig. 22), demonstrating improved robustness to the $E \to 0$ degeneracy that affects the baseline Chinchilla form.

Across all 299 pairs, the baseline 5-parameter model can show lower median CV for exponents but higher for amplitudes, though median statistics hide heavy tails of catastrophic failures. The 7-parameter model reduces certain catastrophic failures by relaxing pathological parameter coupling through scale-modulation terms, but can introduce instability in the modulation exponents and may worsen overall prediction uncertainty in aggregate (Fig. 23).

### G.10 CONCLUSIONS

We evaluated three variants: (i) the baseline 5-parameter Chinchilla model, (ii) a 5-parameter model with distance-dependent $E$-regularization, and (iii) a 7-parameter model with scale-dependent am-

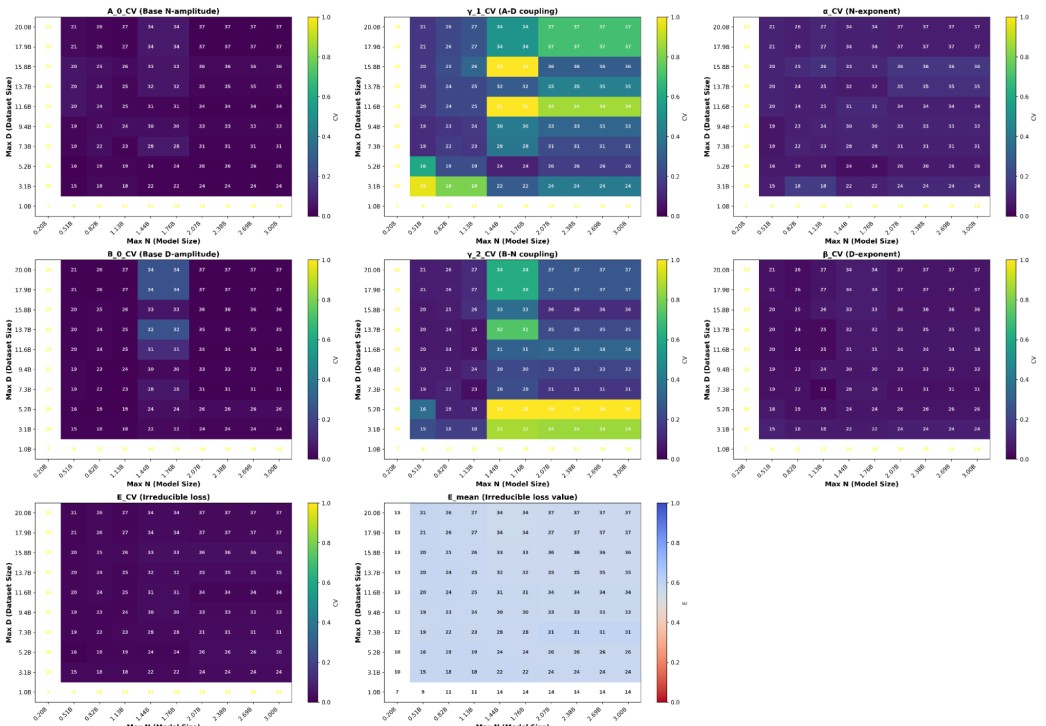

Figure 21: **Parameter uncertainty vs. coverage for the 7-parameter model (in-distribution; orig→val).**

plitudes. We select the $E$-regularized 5-parameter model for its simplicity and strong performance: out-of-bag $R^2 = 0.92$, 84% of pairs with prediction-CV $< 1$, and mean prediction-CV $= 0.45$.

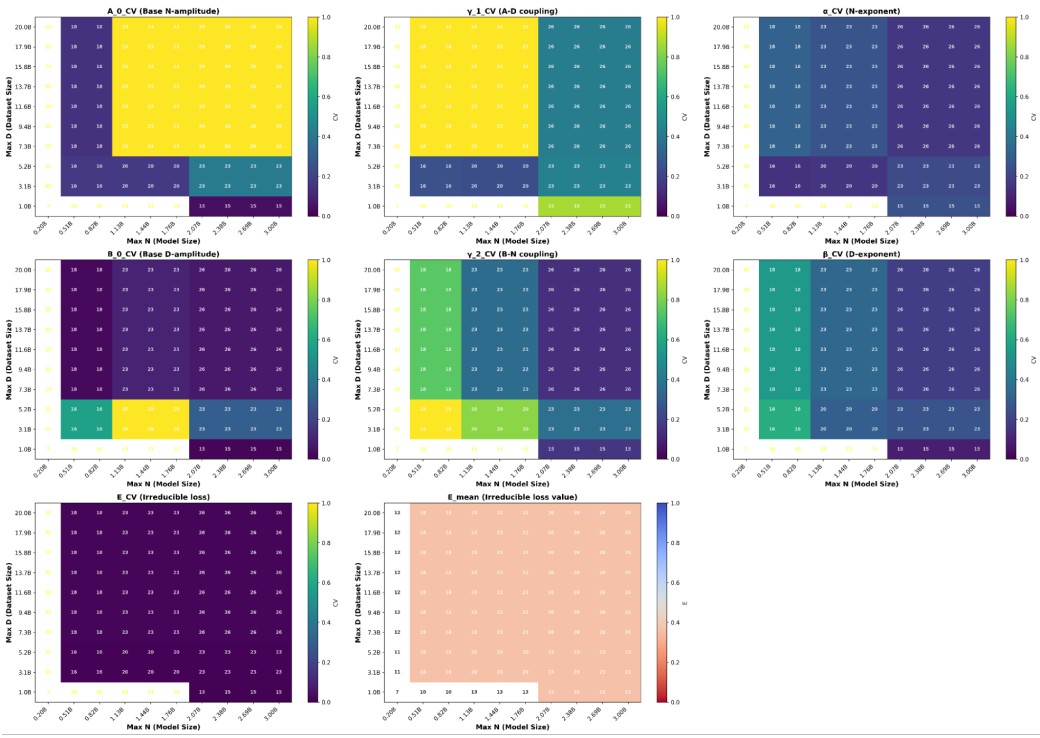

Figure 22: **Parameter uncertainty vs. coverage for the 7-parameter model (distribution shift; aesthetic→math).**

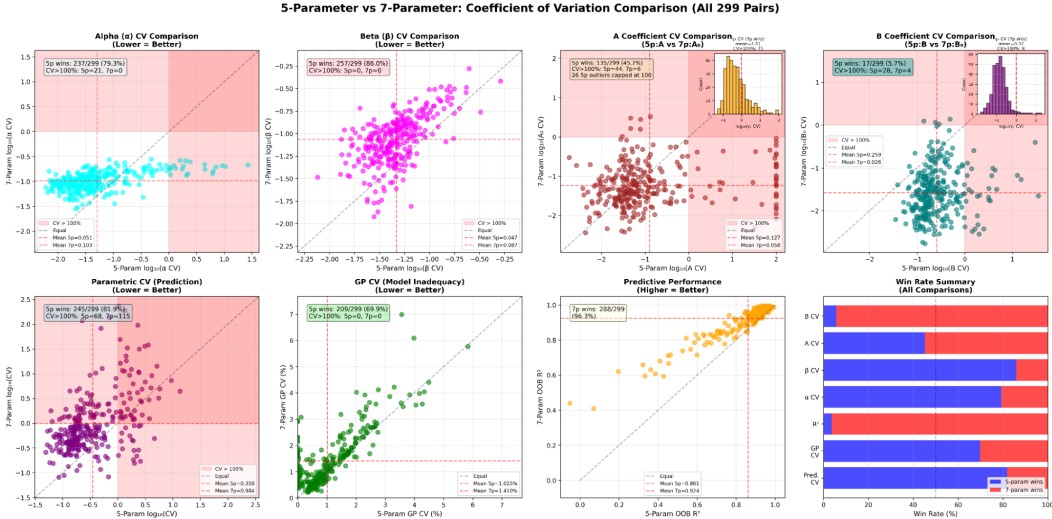

Figure 23: **Parameter uncertainty and predictive performance: baseline 5-parameter vs. 7-parameter model.** Top: parameter CVs for $\alpha, \beta, \gamma_1, \gamma_2, A, B$. Bottom: prediction CV from parameter uncertainty, GP CV, out-of-bag $R^2$, and win rates. Red shading indicates catastrophic failures (CV $> 100\%$).

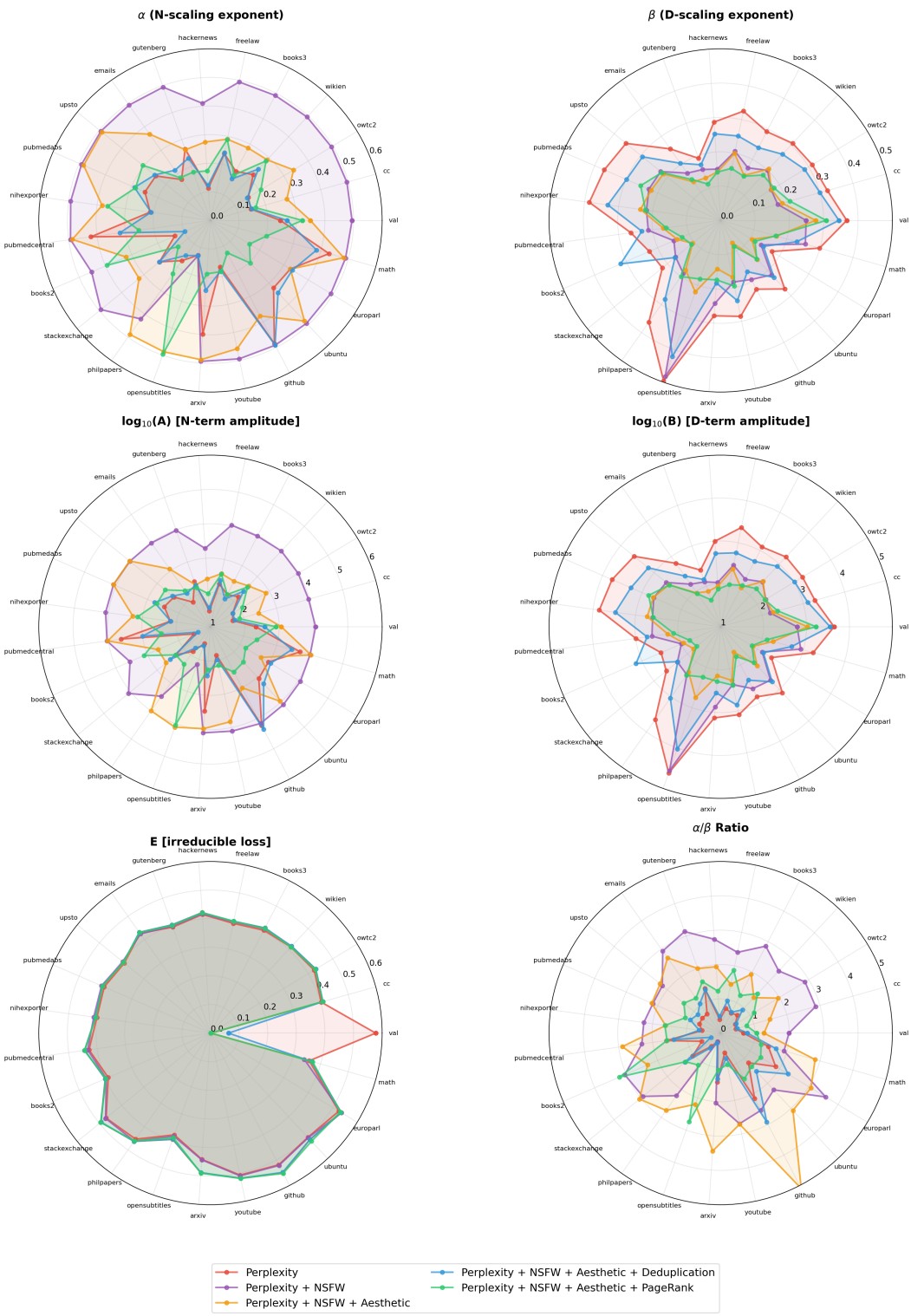

Figure 24: **Scaling-Law Parameter Shifts Persist Under Stabilized Fitting**. Scaling-law parameters ($\alpha$, $\beta$, $\log_{10} A$, $\log_{10} B$, $E$) and the $\alpha/\beta$ ratio estimated with entropy-based $E$ regularization, the stable scaling-law recipe. While parameter estimates vary across evaluation domains due to train–test distribution shift, the consistent separation between filter-specific curves across validation sets indicates that data-quality interventions induce systematic shifts in scaling-law parameters that persist even under stabilized fitting. This suggests that the observed changes in exponents and coefficients are not solely artifacts of scaling-law fit instability in the Chinchilla form, but reflect genuine differences in the effective training distribution. Amplitude parameters $A$ and $B$ exhibit the largest filter-driven variation across domains.