# OpenReview forum: "How Text Quality Interventions Reshape Neural Scaling Laws for LLMs: Empirical Study"
_ICLR.cc/2026/Conference — ICLR 2026 Poster_

### Official Review · Reviewer_922u · 2025-10-31

**Soundness:** 3
**Presentation:** 3
**Contribution:** 3
**Rating:** 6
**Confidence:** 4

**Summary:**

training LLM's is very cost intensive, i.p. w.r.t. required number of tokens. recent pretraining dataset filters battle claiming faster downstream performance adoption. most prominent are heuristic quality filters or synthetic data augmentation. this paper is a massive evaluation on several models and datasets , claiming in fact that correlations between quality interventions and scaling law parameters can be found.

**Strengths:**

- trained on lots of models to provide robust empirical evidence that data quality influences the exponents, not just coefficients of scaling laws
- very relevant for future vvery costly model trainings
- they publish a QualityPajama benchmark suite, again very relevant

**Weaknesses:**

W1 it empirically measures some correlation to exponents, but with little theoretical explanations or evidence. there is only that vague dismissal of zipfian theory

W2 unfortunately i don't see a strong correlation in the quality filter domains and to validations sets, while it appears rather clear / obvious to hold in the synthetic LLM data augmentation. i therefore think it should be phrased that way from the very beginning? renders interpretation of the claimed findings a bit weird.

W3 for the quality filters it seems strange and outdated filters are used. recently most significant are bert-style quality filters, however the authors rely on rather old pagerank?

(W4 the only chosen metric the authors compare to is perplexity, but i also wouldn't know what to do otherwise in this scale.)

**Questions:**

addressing aboves weaknesses

---

### Official Review · Reviewer_TyiW · 2025-10-31

**Soundness:** 3
**Presentation:** 3
**Contribution:** 3
**Rating:** 6
**Confidence:** 3

**Summary:**

This paper asks how interventions on data quality shape LLM scaling behavior. They introduce a benchmark named QualityPajama that consists of 23 datasets derived from CommonCrawl and train >2K models on the datasets. Analyzing the scaling laws, they find that different data quality interventions lead to different relations between parameters, and that picking the optimal (given compute) token-to-parameter ratio depends on the data quality intervention.

**Strengths:**

- Experiments are extensive, covering 2000 models across a large range of model sizes.
- Some of the key findings, for example that the intervention of deduplication can yield large compute savings, have practical downstream utility for people training models with limited compute.

**Weaknesses:**

The paper lacks a theoretical framework through which to understand the extensive empirical results. In particular, the paper would be greatly strengthened if it introduced a new scaling law to relate data quality interventions to other parameters and to explain the empirical findings.

**Questions:**

Formatting notes:
- Line 39: Citations are formatted weirdly
- Line 87: Missing space
- Line 123: period should be comma
- Line 178: formatting of paragraph is weird and inconsistent

---

### Official Review · Reviewer_Mu49 · 2025-11-01

**Soundness:** 3
**Presentation:** 3
**Contribution:** 3
**Rating:** 8
**Confidence:** 2

**Summary:**

The paper studies the effect of data quality on LLM scaling law. The authors start from the RedPajama dataset, consider 14 Heuristic-based data quality interventions and 9 synthetic ones, and applied them in combination to induce various quality-intervened training datasets. For each such quality-invented training dataset, the authors use it to train various sized LLaMA models and evaluate their scaling law. Many findings on the specific effect of different data quality interventions on scaling laws are discussed.

**Strengths:**

1. The paper is written excellent, easy to read and follow.
2. To my knowledge, it is the first work that analyses how specific dataset filtering methods as quality inventions affect specific coefficients in the scaling law. And the authors consider all coefficients rather than just the dataset size coefficient D.
3. Many findings that I expect to be practically helpful in guiding training data preparation as an integral part of model training are discussed.

**Weaknesses:**

1. a small point for improvement is that for Figure 3, 4, 5, the legends block significant portion of the figures (especially Figure 4) and makes the figure less readable. I suggest the authors improve the presentation of these figures so that legends do not block figures.

2. The paper focuses on the classic form of the scaling law and it mentions in related work that Chang et al. (2024) and Muennighoff et al. (2023) introduces the effective dataset size D'. And there might be other extended forms of the scaling laws. Could the authors consider re-run the fitting and analysis with these extended forms beside the classic one? I believe it will be very helpful because (1) it will be an ablation study strengthening the validity of the findings and (2) these extended forms might have better goodness of fit because they have more variables in the system and might fit the scaling law better.

**Questions:**

no questions

---

### Official Review · Reviewer_1zGS · 2025-11-01

**Soundness:** 3
**Presentation:** 3
**Contribution:** 4
**Rating:** 6
**Confidence:** 3

**Summary:**

This paper presents a large scale empirical study of how data quality interventions (deduplication, heuristic filtering and LLM-guided rewriting) affect neural scaling laws in large language model training. They use 23 systematically curated datasets to train over 2,000 models to measure how text quality interventions reshape scaling law components (coefficients A, B and exponents α, β). A key finding is that data quality interventions shift both exponents and coefficients, fundamentally changing scaling dynamics in ways not predicted by existing theory.

**Strengths:**

1. The paper addresses a practically important question that has been largely neglected despite being central to LLM development.
2. Over 2000 model training runs across multiple scales (100M–8B parameters) and token ranges (100M–200B) provides strong empirical support.
3. Clear demonstration that optimal token-to-parameter ratios vary significantly  across interventions at the same compute scale.
4. Well written paper with clear progression from dataset design → empirical findings → interpretations → implications.

**Weaknesses:**

1. While the paper shows that data interventions shift all scaling law components, it provides limited mechanistic understanding of WHY this happens. The discussion of Zipfian theory (Appendix G) shows weak correlations but doesn't propose alternative explanations. There is critique of existing theory (data manifold, Zipfian distribution) but it is somewhat superficial. Would have been interesting to see more discussion here.
2. Filters are applied sequentially, but the order matters (acknowledged in Section 4). The paper doesn't fully disentangle whether observed component shifts are due to the filter itself or the interaction order.
3. The 0.3-0.5 Spearman correlation for heuristic filters is interesting but not fully explained. Why do natural filters show lower consistency across validation sets than synthetic filters (0.91 for B)?
4. The paper makes several claims about deduplication efficiency that appear to use different baselines or comparison points:
  - "Exact deduplication reduces data volume to 83% of original yet yields a 100× gain in compute efficiency" (L431-432)
  - "Fuzzy deduplication (0.7) requires approximately 3× less compute than 0.9, 10× less than exact deduplication, and 300× less than no deduplication" (L459-461)
  - These statements are difficult to compare together. If exact dedup achieves 100× gain, is fuzzy 0.7 achieveing "300× less” than the baseline or exact dedup? What am I missing here?

5. All experiments use llama3-style transformer. Results may not generalize to other architectures (mixture-of-experts, alternative attention mechanisms, etc.). This doesn’t require more experiments, but limitation discussion could be clearer on this.
6. The claim that "deduplication could expand the data manifold" is counterintuitive and under-explained. Deduplication removes redundant documents. If a document is a duplicate, removing it changes weighting of the remaining documents but how does this expand the manifold? This needs clarification or revision.

## Minor
- The paper uses \citet in several places where \citep would be more appropriate. Many claims about what "prior work overlooks" (e.g., L58 "what prior work overlooks other components?") lack supporting citations. Strengthen argumentative claims with explicit references.
- L116 The statement that α, β, etc "are constants" is not helpful. What do these constants model/support? Provide intuitive descriptions (e.g., "α quantifies model scaling efficiency").
- Consider inverting the right column of figures so that "higher is always better" visually. This would make the figure easier to interpret as a whole.
- “LLM” is not abbreviated on first use in the intro. It should be defined on first use then re-used.

**Questions:**

1. How sensitive are the scaling law components to the curve-fitting method? Have you tested non-parametric approaches or Bayesian inference?
2. The paper uses single-epoch training. How would results change with data repetition? Does the data manifold interpretation change?
3. For synthetic data generation, did you experiment with different generator models or temperature settings?
4. Can you provide theoretical intuition for why data interventions shift both exponents and coefficients rather than just one?

---

### Author Response · Authors · 2025-12-03
**Author Response to All Reviews and Summary for AC**

We thank all reviewers for their constructive feedback and their recognition of our work's significance. Reviewer 1 highlights that our paper addresses a "practically important" yet "largely overlooked question which is central to LLM development. Reviewer 2 notes this is "the first work to systematically analyze how quality interventions influence scaling-law behavior." Reviewer 3 emphasizes the "practical utility" of our findings for practitioners training models under limited compute budgets. Reviewer 4 finds the work highly relevant given the growing cost of large-scale model training.

This paper offers a new perspective on how data-quality interventions influence neural scaling laws. While prior work noted that architecture can affect coefficients, our results are the first to show that data interventions shift both coefficients and exponents, fundamentally altering the scaling relationship in ways not predicted by existing theory.  During the rebuttal period, we conducted additional experiments and added **Appendix I: A Step-by-Step Guide to a Stable Scaling-Law Form**, documenting failure modes of the Chinchilla form and our stabilized solution. We believe our systematic framework and new stabilized scaling law formulation provide a strong foundation for future theoretical and empirical work in this area.

Below we address the key questions raised.

---

### Q1. Parameter Sensitivity to Fitting Methods (R1)

We compared the MLE bootstrap approach (original Chinchilla approach) against MCMC Bayesian inference. We found that the fitted uncertainties (coefficients of variation) are remarkably tighter for the MLE bootstrap approach. MCMC produces extremely large or unstable posterior variances for A, B, and E.

Comparing the average estimates from MCMC and MLE, we find their estimations are highly correlated (Pearson r):
- **β** (data exponent): r = 0.961
- **B** (data prefactor): r = 0.956
- **α** (model exponent): r = 0.724
- **A** (model prefactor): r = 0.743

The only parameter with poor agreement is **E** (asymptotic loss): r = 0.274. For E, 59% of (training, validation) pairs collapse to zero under MLE, whereas MCMC estimates E > 0 but with large uncertainty (E = 0.64 ± 0.56).

---

### Q2. Theoretical Intuitions (R1, R4)

Data interventions systematically influence the exponents and the coefficients. While we do not have a formal proof, the intuition is straightforward once we interpret the components of the scaling law:

- The **exponents** (α, β) capture how efficiently loss improves with scale, i.e., the curvature of the scaling law.
- The **coefficients** (A, B) and E capture the baseline difficulty of the task, i.e., the vertical position of the curve.

Data quality interventions affect both aspects of the learning problem. For example, less redundant data lowers the baseline loss (coefficient shift) and changes how much benefit comes from scaling (exponent shift). Thus, both components move because data quality simultaneously affects where the curve starts and how quickly it improves.

---

### Q3. Choice of Quality Filters (R4)

Our filters are deliberately chosen to represent **common denominators** across successful data recipes (C4, Dolma, RedPajama, RefinedWeb, FineWeb). These include:
- Perplexity-based filtering
- NSFW filtering
- Format-based filtering
- Grammar-based filtering
- Deduplication
- PageRank score (less explored but informative)

**Why these filters?** They allow **controlled, comparable interventions** where the research question is whether data interventions affect scaling law components—not which specific filter is "best." While absolute values for each parameter may vary across filters, the **core finding is robust**: data quality interventions systematically influence *all* components of the power law (exponents, coefficients, and asymptotic loss).

We agree that exploring cutting-edge filtering techniques is valuable future work, but our current filter set provides a strong foundation for understanding the *structural* relationship between data quality and scaling behavior.

---

### Q4. How Does Epoching/Data Repetition Influence Scaling Laws? (R1, R2)

We can view epoching as the **inverse of deduplication**. Empirically we observed that:
**Deduplication** increases α (stronger gains from model scaling)  and slightly reduces β (weaker gains from additional data). Hence, we expect **Epoching/repetition** lowers α (weaker gains from model scaling) and slightly raises β when measured against raw token count. Repeated tokens inflate the data count D without adding new information, which:
1. Reduces the effective support for learning (hurts model scaling efficiency → lower α)
2. Creates an illusion of "more data" when measured in raw tokens (inflates β artificially)

---

### Meta-Review · Area_Chair_yndn · 2026-01-07

**Summary:**

The paper analyses how different data quality interventions (e.g. deduplication, filtering, LLM rewriting) affect scaling laws. The paper trains over 2k models to show how these interventions on data affect different scaling law components.

All reviewers are positively leaning and the paper addresses an important question with high value for LLM training.

The authors also provide a rebuttal to address issues and have added a new Appendix to further strengthen the paper. While the point of there not being theory to explain why the interventions affect things stands, that this doesn’t diminish the contribution around showing the empirical phenomena and how data quality interventions affect scaling in different ways. This in itself is a valuable contribution for the community

Recommendation: On this basis I vote to accept the paper. The authors are encouraged to address the issues around figure readability raised by the reviewers in the camera ready.

**Reviewer Concerns:**

see "Summary" text

**Reviewer Scores:**

see "Summary" text

---

### Decision · Program_Chairs · 2026-01-26

Accept (Poster)